# RewardSDS: Aligning Score Distillation via Reward-Weighted Sampling

"*A penguin with a brown bag in the snow.*"

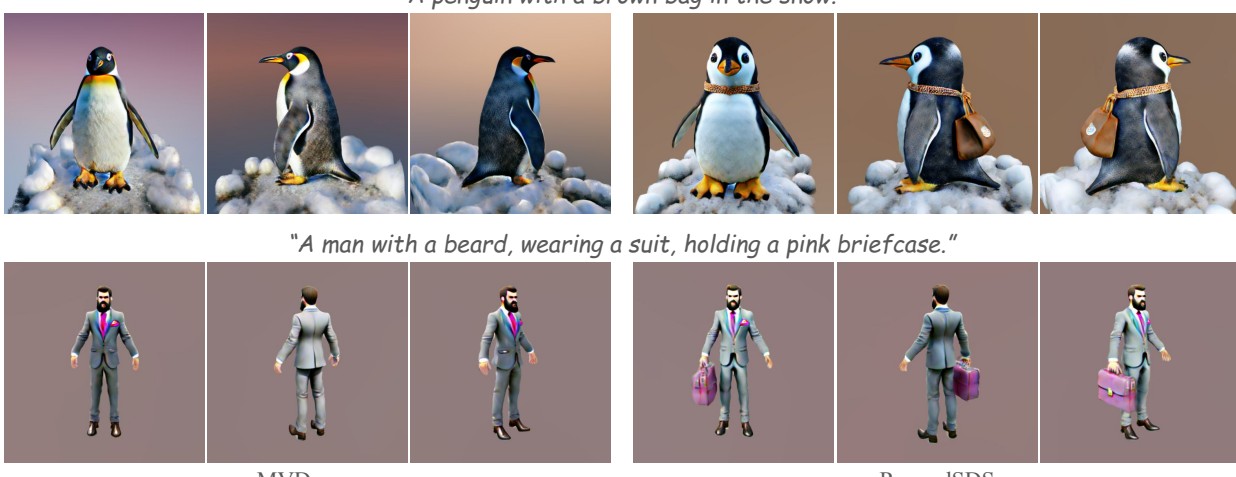

"*A man with a beard, wearing a suit, holding a pink briefcase.*"

MVDream            RewardSDS

Figure 1: RewardSDS is a plug-and-play score-distillation enhancement that enables reward-aligned generation across tasks and SDS variants. Here we apply it to MVDream Shi et al. (2024) for text-to-3D generation.

## Abstract

Score Distillation Sampling (SDS) has emerged as a highly effective technique for leveraging 2D diffusion priors for a diverse set of tasks such as text-to-3D generation. While powerful, SDS still struggles to achieve fine-grained alignment, particularly in capturing the semantic details or perceptual qualities specified by the target prompt. To overcome this limitation, we introduce *RewardSDS*, a novel approach that weights noise samples based on the alignment scores of a reward model, producing a weighted SDS loss. This loss prioritizes gradients from noise samples that yield aligned high-reward output, enabling controllable alignment without requiring differentiable reward models. Our approach is broadly applicable and can be applied to diverse methods extending SDS. In particular, we also demonstrate its applicability to Variational Score Distillation (VSD) by introducing RewardVSD. We evaluate RewardSDS on text-to-image, 2D editing, and text-to-3D generation tasks, demonstrating a significant improvement over SDS and subsequent baselines on a diverse set of metrics measuring generation quality and alignment goals.

## 1 Introduction

Diffusion models have shown remarkable success in generating high-fidelity and diverse images (Rombach et al., 2022a; Saharia et al., 2022; Ramesh et al., 2022) and videos (Ho et al., 2022; Singer et al., 2022; Blattmann et al., 2023). However, their success often hinges on the availability of large-scale datasets, a requirement that poses a significant challenge in modalities like 3D content generation. In these data-scarce scenarios, diffusion models are often used as priors to guide the optimization of generative models without requiring paired supervision. A key framework for this is Score Distillation Sampling (SDS) (Poole et al.,

2022; Wang et al., 2023a), which distills the guidance of a 2D diffusion model into a 3D representation by optimizing a score-based objective on rendered images.

While SDS and its variants have enabled significant progress, achieving fine-grained controllability, remains challenging. A recent effort, DreamReward Ye et al. (2024), explores integrating reward signals into SDS by training a differentiable reward model on multi-view 3D data and using its gradients to adjust the SDS objective. While effective, this approach requires costly multi-view annotations and differentiable reward models, limiting its generality and its ability to align outputs with diverse user-defined objectives.

Motivated by recent advances in reward-guided sampling and scaling-based alignment strategies in LLMs Brown et al. (2024); Snell et al. (2024), and diffusion models (Wallace et al., 2024; Karthik et al., 2023; Liu et al., 2024; Zhou et al., 2024; Ma et al., 2025), we revisit the optimization dynamics of SDS itself. SDS-based methods compute gradients from a pre-trained diffusion model using noisy samples, implicitly assuming that all noise samples contribute equally. However, some noise samples may correspond to high-reward regions in the output space, while others may lead to low-reward regions. Motivated by these works, we ask: *How can one effectively harness reward-based sample selection to align outputs produced by score distillation?*

To address this question, we propose a novel adaptation of SDS, called RewardSDS. We begin by rendering an image $x$ from an underlying model $\theta$. We then draw $N$ noises corresponding to timestep $t$ from a Gaussian distribution, resulting in noisy samples $x_t^1, \ldots x_t^N$. Each noisy sample $x_t^i$ is assigned an *alignment score*. This score is obtained by first denoising $x_t^i$ using the diffusion model, and then feeding the denoised sample into a given reward model to obtain a reward value, which we term the "alignment score". The overall loss is then given by a *weighted sum* of the SDS losses computed for individual noisy samples $x_t^1, \ldots x_t^N$, where the weight is derived from the alignment score. This simple yet general modification enables SDS-based methods to align their outputs more closely with desired reward functions, without requiring differentiable rewards or multi-view supervision.

RewardSDS is general, modular, and computationally scalable. It can be seamlessly integrated into existing SDS-based pipelines, such as Variational Score Distillation (VSD) (Wang et al., 2023b), MVDream (Shi et al., 2024), and others, without requiring significant changes. In particular, we demonstrate its compatibility with VSD by introducing RewardVSD, which applies our reward-weighted formulation within a variational optimization framework. By leveraging additional inference-time compute, RewardSDS improves both semantic and perceptual alignment, enabling a controllable trade-off between computational cost and generation quality.

We evaluate RewardSDS across multiple SDS-based methods and tasks, including text-to-image generation, image editing, and text-to-3D generation. Across all evaluated settings, RewardSDS consistently improves upon these methods, producing visually sharper, more coherent, and better-aligned results. These improvements demonstrate that reward-guided noise weighting enhances overall generation quality. Finally, our ablation studies reveal a clear compute–performance trade-off: even modest additional compute leads to significant improvements, while larger compute budgets further enhance generation quality.

To summarize, our contributions are as follows:

- We propose RewardSDS, a reward-weighted extension of Score Distillation Sampling that aligns diffusion-based optimization with arbitrary reward functions.
- RewardSDS is a plug-and-play approach, seamlessly integrating into existing SDS-based frameworks such as VSD and MVDream without architectural changes.
- We provide a comprehensive analysis showing a clear compute–performance trade-off, with consistent gains in alignment and visual quality across diverse tasks.

## 2 Related Work

**Diffusion Models Alignment**    Fine-tuning diffusion models for alignment with human preferences has been extensively explored through methods such as reinforcement learning, which optimizes reward signals Black et al. (2023), direct gradient backpropagation from reward functions Xu et al. (2023), direct

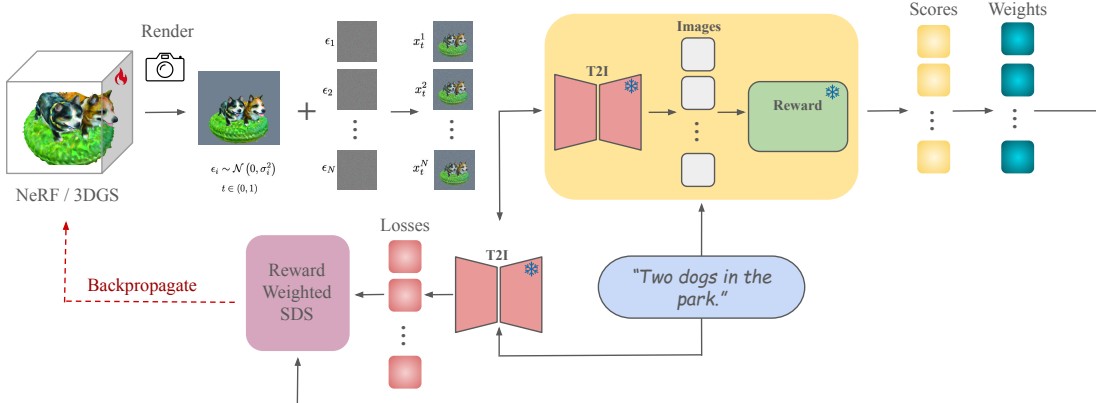

Figure 2: **RewardSDS illustration.** An image is first rendered from a given view and $N$ random noises are applied (at a given timestep). The noisy images are then scored by denoising them and applying a reward model on the output. These scores are then mapped to corresponding weights, which are used to weigh the contribution of each noisy sample in score distillation.

preference optimization based on diffusion likelihood Wallace et al. (2024), and stochastic optimal control frameworks Domingo-Enrich et al. (2024).

Sample selection and optimization have also emerged as an important paradigm for improving diffusion model sample quality and alignment at test time. Some methods focus on metric-guided selection, using random search guided by a reward model Karthik et al. (2023); Liu et al. (2024), similarity to references Tang et al. (2024); Samuel et al. (2024), or approximating "good" noise distributions with neural networks Ahn et al. (2024); Zhou et al. (2024). Unlike these works, we consider alignment in the context of score distillation, where the impact of noise selection on gradient updates is fundamentally different.

**Score Distillation Sampling**     Score Distillation Sampling (SDS) Poole et al. (2022); Wang et al. (2023a) has emerged as an essential technique for text-to-3D by leveraging the 2D prior from text-to-image diffusion models to optimize rendering parameters, such as NeRF Mildenhall et al. (2020). However, the original SDS formulation exhibits limitations, notably artifacts like over-saturation and over-smoothing.

Several works have aimed to improve the core distillation process with different strategies. Variational Score Distillation (VSD) Wang et al. (2023b) reformulates SDS as particle-based variational inference. Classifier Score Distillation Yu et al. (2023) re-evaluates the role of classifier-free guidance in score distillation. Dream-Flow Lee et al. (2024) suggests using a predetermined timestep schedule for distillation. JointDreamer Jiang et al. (2024) further improves multi-view consistency by using view-aware models within a joint distillation framework. SDS-Bridge McAllister et al. (2025) presents a unified generalized score distillation framework. Delta Denoising Score (DDS) Hertz et al. (2023a) and Posterior Denoising Score Koo et al. (2024) adapt SDS for image editing by modifying the gradient direction to better preserve input image details during editing. *Beyond these core algorithmic improvements*, research has also explored diverse applications of SDS such as generating SVG graphics Iluz et al. (2023); Jain et al. (2023), sketches Xing et al. (2023), textures Li et al. (2024); Metzer et al. (2023), typography Iluz et al. (2023), and dynamic 4D scenes Bahmani et al. (2024); Singer et al. (2023). Importantly, these advancements are orthogonal to our reward-based approach and offer potential for synergistic combination. Liang et al. (2024); Lukoianov et al. (2024) proposed to replace the randomly sampled noise at each SDS step with trajectories that are consistent with previously predicted noise by using DDIM Inversion Mokady et al. (2023); Miyake et al. (2025). While offering a type of noise selection, as we do in our method, our work enables this choice according to the desired reward-model alignment, offering a diverse set of solutions.

In the context of SDS alignment, DreamReward Ye et al. (2024) has recently proposed to align SDS by training a reward model to score a set of multi-view images. They then use this model to adjust the SDS score provided by the underlying 2D diffusion model. Our method differs in two main aspects: (1). Our method can be aligned with any pretrained reward model, specifically a reward model trained to align

2D images as well as non-differentiable reward models. In contrast, DreamReward require the expensive annotation of scores for multi-view data. (2). DreamReward still perform an expectation over all noises corresponding to a given timestep. Our method, on the other hand, weighs different noises according to their alignment score. Thus, our contribution offers a different way to integrate reward information directly into the SDS optimization process.

## 3 Method

We begin by outlining related background, specifically score distillation sampling (SDS) and variational score distillation (VSD). We then present our approach and show how it can be applied to both standard SDS and its extensions, using VSD as a representative example (while also evaluating its plug-and-play nature with additional variants in Sec. 4, and Appendix A.3). An illustration of our method is provided in Fig. 2.

**Score Distillation Sampling (SDS)**   Given a clean image $x_0$ and a text prompt $y$, diffusion models are trained to predict the noise $\epsilon_t$ added at timestep $t$ to a noisy image $x_t$. This training objective can be expressed as:

$$L(x_0) = \mathbb{E}_{t \sim U(0,1), \epsilon_t \sim N(0,I)} \left[ w(t) \| \epsilon_\phi(x_t, y, t) - \epsilon_t \|_2^2 \right] \tag{1}$$

where $\epsilon_\phi(x_t, y, t)$ is the noise predictor, $w(t)$ is a weighting function, and $x_t$ is obtained through the forward diffusion process:

$$x_t = \alpha_t x_0 + \sigma_t \epsilon_t, \quad \epsilon_t \sim N(0, I) \tag{2}$$

where $\alpha_t, \sigma_t$ are hyperparameters, such that $\sigma_t^2 + \alpha_t^2 = 1$, and $\sigma_t$ gradually increases from 0 to 1.

In SDS, when $x_0 = g(\theta)$ is a rendered image from a differentiable generator $g$ parameterized by $\theta$, the parameters $\theta$ are updated by backpropagating the gradient of the loss:

$$\nabla_\theta L_{SDS}(x_0 = g(\theta)) = \mathbb{E}_{t, \epsilon_t} \left[ w(t)(\epsilon_\phi(x_t, y, t) - \epsilon_t) \frac{\partial x_0}{\partial \theta} \right] \tag{3}$$

This gradient update guides the parameters $\theta$ such that the rendered images $g(\theta)$ increasingly resemble samples from the text-conditioned 2D diffusion model. Intuitively, SDS perturbs a rendered image $x_0$ by adding Gaussian noise to obtain $x_t$, then uses the pre-trained diffusion model to predict the noise $\epsilon_\phi(x_t, y, t)$ that should be present in $x_t$ to move it towards the real image distribution. The difference between the predicted noise and the added noise, weighted by $w(t)$ and backpropagated through $\frac{\partial x_0}{\partial \theta}$, provides an approximation of the gradient for optimizing $\theta$.

**Variational Score Distillation (VSD)**   Despite its effectiveness, SDS often suffers from issues like over-saturation and over-smoothing. To address these limitations and improve generation quality and diversity, Variational Score Distillation (VSD) Wang et al. (2023b) was proposed. VSD reformulates SDS within a particle-based variational inference framework. Instead of optimizing a single 3D scene representation, VSD optimizes a distribution $\mu(\theta|y)$ over scene parameters by introducing multiple particles $\{\theta_i\}_{i=1}^n$. The VSD objective is defined as:

$$\mu^* = \arg\min_\mu \mathbb{E}_t \left[ \frac{\sigma_t}{\alpha_t} \omega(t) D_{KL}(q_t^\mu(x_t|y) \| p_t(x_t|y)) \right] \tag{4}$$

where $q_t^\mu(x_t|y)$ is the distribution of rendered images from the particle set, and $p_t(x_t|y)$ is the target distribution from the pre-trained diffusion model. To solve this optimization, VSD fine-tunes an additional model, $\epsilon_\phi$, using LoRA, to estimate the score of the proxy distribution $q_t^\mu(x_t|y)$. That is, $\epsilon_\phi$ is finetuned on rendered images by $\{\theta^{(i)}\}_{i=1}^K$ with the standard diffusion objective:

$$\mathbb{E}_{t, \epsilon_t} \left[ w(t) \| \epsilon_\phi(x_t, y, t) - \epsilon_t \|_2^2 \right] \tag{5}$$

The gradient for each particle $\theta_i$ in VSD, $\nabla_{\theta_i} L_{VSD}(\theta_i)$ is then computed as:

$$\mathbb{E}_{t, \epsilon_t} \left[ w(t)(\epsilon_{pretrain}(x_t, y, t) - \epsilon_\phi(x_t, y, t)) \frac{\partial g(\theta_i)}{\partial \theta_i} \right] \tag{6}$$

where $\epsilon_{pretrain}$ is the original pre-trained diffusion denoiser, and $\epsilon_\phi$ is the fine-tuned score estimator.

**RewardSDS**  Our core idea is to incorporate a reward model $R$ to guide the SDS optimization by prioritizing noise samples that are more likely to produce high-quality, aligned outputs. To this end, we rank a set of $N$ noise samples $\{\epsilon_t^{(i)}\}_{i=1}^N$ drawn at each iteration based on the reward scores they induce in the rendered image. Specifically, for a given timestep $t$ and a set of noise samples $\{\epsilon_t^{(i)}\}_{i=1}^N \sim N(0, I)$, we generate a set of noisy images $\{x_t^{(i)}\}_{i=1}^N$ using Eq. 2:

$$x_t^{(i)} = \alpha_t x_0 + \sigma_t \epsilon_t^{(i)}, \quad i = 1, 2, ..., N \tag{7}$$

where $x_0 = g(\theta)$ is the rendered image. We then evaluate each noisy image $x_t^{(i)}$ by first denoising it and then using a reward model $R$ to obtain a set of reward scores $\{r^{(i)}\}_{i=1}^N$, where $r^{(i)} = R(x_0^{'(i)})$. $R$ may optionally accept a text input.

Based on these reward scores, the SDS loss for each noise sample $\epsilon_t^{(i)}$ is then weighted by a factor $w^{(i)}$ that is derived from $r^{(i)}$ (see Sec. 4 for details).

The Reward SDS loss is then computed as a weighted sum of the individual SDS losses, where $\{w^{(i)}\}_{i=1}^N$ are the weights assigned based on the reward ranking, and $w(t)$ is the standard SDS weighting function. The gradient for updating the parameters $\theta$, $\nabla_\theta L_{R-SDS}(x_0 = g(\theta))$, is then:

$$\mathbb{E}_t \left[ \frac{1}{N} \sum_{i=1}^N w^{(i)} \left[ w(t)(\epsilon_\phi(x_t^{(i)}, y, t) - \epsilon_t^{(i)}) \frac{\partial x_0}{\partial \theta} \right] \right] \tag{8}$$

**RewardVSD**  In RewardVSD, for each particle $\theta_i$ in the particle set $\{\theta_i\}_{i=1}^K$, we sample a set of $N$ noise samples $\{\epsilon_t^{(i,j)}\}_{j=1}^N$. For each noise sample $\epsilon_t^{(i,j)}$, we compute the noisy image $x_t^{(i,j)}$ and its corresponding reward score $r^{(i,j)} = R(x_0^{'(i,j)})$. We then rank the noise samples $\{\epsilon_t^{(i,j)}\}_{j=1}^N$ based on their reward scores $\{r^{(i,j)}\}_{j=1}^N$ and assign weights $\{w^{(i,j)}\}_{j=1}^N$ accordingly. The gradient for updating particle $\theta_i$, $\nabla_{\theta_i} L_{R-VSD}(\theta_i)$, is:

$$\mathbb{E}_t \left[ \frac{1}{N} \sum_{j=1}^N w^{(i,j)} \left[ w(t)(\epsilon_{pretrain}(x_t^{(i,j)}, y, t) - \epsilon_\phi(x_t^{(i,j)}, y, t)) \frac{\partial g(\theta_i)}{\partial \theta_i} \right] \right] \tag{9}$$

For completeness, Appendix C provides additional theoretical analysis of RewardSDS, including its formal connection to preference-based optimization, and an empirical examination of reward variance during optimization.

## 4  Experiments

We begin by evaluating our method on zero-shot text-to-image generation, reporting the effect of different reward models. Second, we evaluate our approach on text-to-3D generation. Next we demonstrate the applicability of our approach for image editing. Lastly, we conduct an extensive ablation study and analyze optimization time vs performance tradeoff of using our method. Detailed implementation information for all experiments is provided in Appendix D.

**Reward models.**  We incorporate CLIPScore Radford et al. (2021) (ViT-L/14), Aesthetic Score Predictor Schuhmann et al. (2022), ImageReward Xu et al. (2023), and Reward3D Ye et al. (2024) as optional reward models. Each model guides optimization based on a different aspect of quality. CLIPScore evaluates text-image alignment by comparing visual and textual features. Aesthetic Score Predictor assesses aesthetic quality, as it is trained to predict human ratings of synthesized images' visual appeal. ImageReward evaluates both alignment and aesthetics, learning general human preferences through a carefully curated annotation pipeline. Reward3D, used for 3D generation, scores a scene based on multiple rendered views, capturing alignment and consistency across different perspectives.

**Metrics.**  As metrics, we utilize the presented reward models. While each model type is optimized with respect to a specific reward model, we evaluate it with respect to all reward models. Additionally, we utilize

Gemini 2.0 Flash [1] as a judge, asking it to score each prompt-image pair on a scale from 1 to 10 based on criteria such as accuracy to the prompt, creativity and originality, visual quality and realism, and consistency and cohesion (we set the temperature to 0 to ensure deterministic scoring.). We refer to this metric as the LLM Grader (LLM-G) and present the average score. The exact prompt used as input for the LLM is sourced from Ma et al. (2025). To further validate our approach, we construct user studies, where we assess the realism of generated outputs and their alignment to the input text.

Table 1: Effect of different reward models on generated outputs using RewardSDS and RewardVSD. Each row represents results obtained by applying our method with a different reward model. The first row corresponds to the baseline SDS or VSD, where no reward model is used. We report scores from three reward models: CLIP, Aesthetic, ImageReward (IG), along with LLM-G.

| Reward Model | CLIP↑ | Aesthetic↑ | IR↑ | LLM-G↑ |
|---|---|---|---|---|
| **RewardSDS** | | | | |
| SDS Baseline | 27.45 | 5.21 | 0.53 | 6.69 |
| CLIP | **28.12** | 5.22 | 0.72 | 6.88 |
| Aesthetic | 27.47 | **5.35** | 0.55 | 6.74 |
| ImageReward | 27.80 | 5.24 | **1.13** | **7.03** |
| **RewardVSD** | | | | |
| VSD Baseline | 27.91 | 5.42 | 0.60 | 6.74 |
| CLIP | **28.98** | 5.49 | 0.78 | 6.92 |
| Aesthetic | 27.30 | **5.69** | 0.64 | 6.83 |
| ImageReward | 28.84 | 5.53 | **1.20** | **7.13** |

Table 2: Comparison of zero-shot text-to-image generation using RewardSDS/RewardVSD with ImageReward compared to SDS/VSD. We evaluate CLIP, Aesthetic, and LLM-G. We also assess image alignment and realism (user study MOS on a scale of 1-5). []

| Method | CLIP↑ | Aesthetic↑ | LLM-G↑ | Align.↑ | Real.↑ |
|---|---|---|---|---|---|
| MS-COCO | | | | | |
| SDS | 27.32 | 5.32 | 6.75 | 2.85 | 2.66 |
| RewardSDS | **27.77** | **5.42** | **7.01** | **3.93** | **3.18** |
| VSD | 27.40 | 5.66 | 6.88 | 3.15 | 2.31 |
| RewardVSD | **27.91** | **5.86** | **7.12** | **4.08** | **2.80** |
| Drawbench | | | | | |
| SDS | 27.19 | 5.34 | 6.70 | 3.18 | 2.34 |
| RewardSDS | **27.95** | **5.55** | **7.02** | **3.48** | **2.66** |
| VSD | 27.65 | 5.70 | 6.73 | 2.85 | 2.34 |
| RewardVSD | **28.16** | **5.77** | **7.09** | **3.82** | **2.95** |

## 4.1 Zero-Shot Text-to-Image Generation

We begin by performing a zero-shot text-to-image generation where we optimize a latent map of size $64 \times 64 \times 4$ (corresponding to an image) in the Stable Diffusion's latent space McAllister et al. (2024); Wang et al. (2023b). As opposed to text-to-3D, this setup isolates the effect of our reward-based score distillation and enables a clearer assessment of its impact.

**Effect of different reward models.** First, we evaluate the effect of different reward models on our approach. We randomly selected 25 prompts from the Drawbench benchmark Saharia et al. (2022), a diverse, general-purpose dataset of text prompts spanning multiple categories. We optimized RewardSDS and RewardVSD, using each of the 2D reward models noted above. Additionally, we report results for the SDS and VSD baselines alone. As shown in Tab. 1, models trained on each of the three reward models effectively improve the baselines. For both RewardSDS and RewardVSD, using the ImageReward model achieves the best overall results, attaining the highest LLM Grader score and the strongest performance on its own reward model. This suggests that ImageReward provides a well-balanced optimization that enhances alignment, aesthetics, and overall human preference. Not surprisingly, in all cases, each reward model excels when evaluated with its corresponding reward model. However, we find that optimizing with respect to one reward model also results in an improved performance with respect to the other reward models.

---

[1] https://ai.google.dev/gemini-api

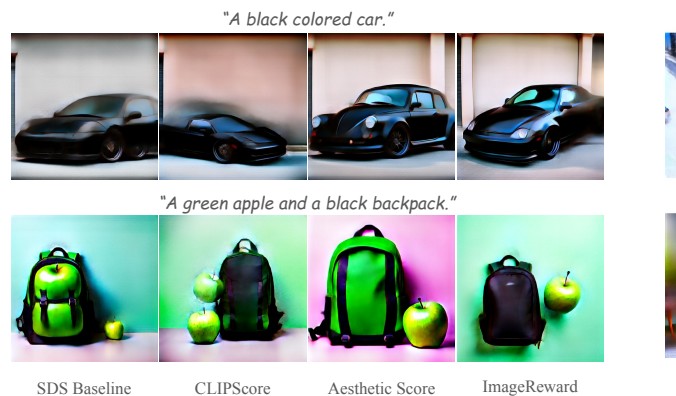

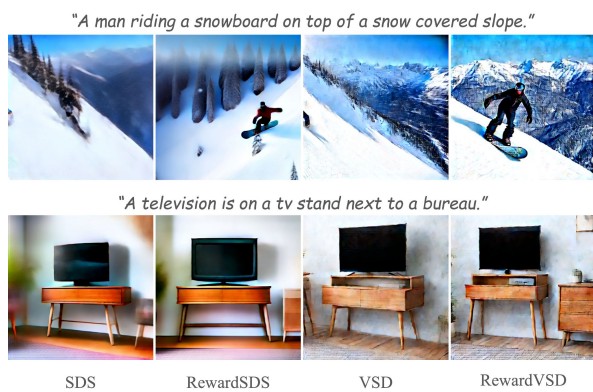

Figure 3: Qualitative comparison of generated outputs using different reward models for RewardSDS and the SDS baseline.

Figure 4: Qualitative comparison of zero-shot text-to-image generation.

Qualitative results showcasing RewardSDS with different reward models, along with the SDS baseline, are presented in Fig.3, while results for RewardVSD with different reward models, along with the VSD baseline, are shown in Appendix B.4. These examples illustrate the effect of integrating each reward model into image generation, along with the baseline.

**Larger-scale comparison to baselines.** To further evaluate our approach, we conduct a larger-scale comparison to baselines. We randomly sample 100 prompts: 50 from MS-COCO Lin et al. (2015) and 50 from Drawbench Saharia et al. (2022). We use ImageReward as the reward model and report results for CLIPScore, Aesthetic Score, and the LLM Grader. To verify that better alignment to the given reward model is not compensated by image realism or by alignment to the input text, we conduct a user study in which 50 users are asked to score from 1 (lowest) to 5 (highest): (1). How realistic is the generated image?, (2). How aligned is the generated image to the input text? For each of the 100 prompts used above, we generated corresponding images using SDS, VSD, RewardSDS and RewardVSD. Users are then presented the image corresponding for each method at random, and asked to rank each image from 1-5 on questions (1)-(2) above. As shown in Tab. 2, our reward-based sampling consistently improves generation performance across all metrics. Qualitative comparisons between RewardSDS, SDS, RewardVSD, and VSD are shown in Fig. 4. To further evaluate generation alignment, we specifically include generations of counting and compositional prompts, showing clear improvements with RewardSDS (Appendix A.4).

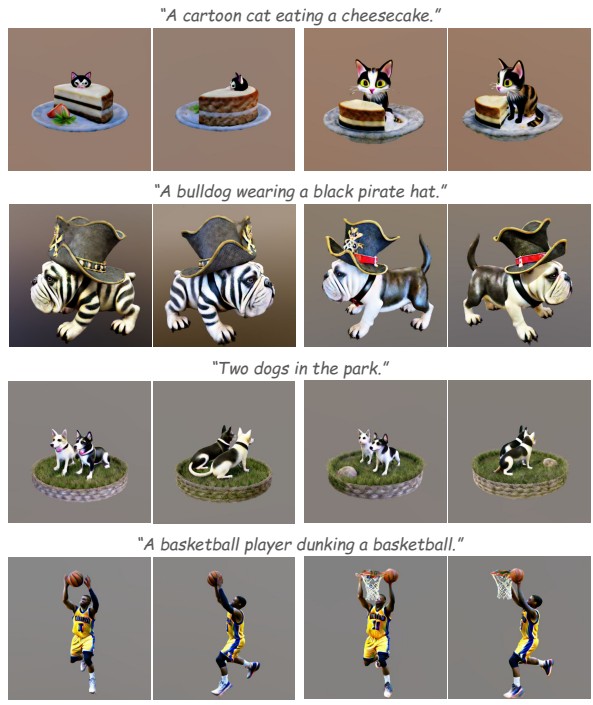

Figure 5: Qualitative comparison of text-to-3D generation based on NeRF, in comparison to MVDream.

## 4.2 Text-to-3D Generation

We now evaluate our approach on text-to-3D generation. Specifically, we follow the state-of-the-art work of MVDream Shi et al. (2024) by first training a multiview diffusion model. We subsequently optimize either a NeRF Mildenhall et al. (2020) or a 3DGS Kerbl et al. (2023) backbone using score distillation on this pretrained diffusion model, following the pipeline of DreamFusion Poole et al. (2022) or DreamGaussian Tang et al. (2023), replacing the use of the SDS loss with RewardSDS. We further evaluate how our reward-based

Table 3: Evaluation of text-to-3D methods, with/without reward-based sampling. Beyond MVDream, we also consider our reward-based sampling when applied with DreamReward, which was trained with a 3D reward model Reward3D (R3D). We assess performance using CLIP, Aesthetic, and LLM-G, as well as alignment and realism through a user study (MOS on a scale of 1-5). Reward models include ImageReward (IR), Aesthetic (A), and Reward3D (R3D).

| Method | Reward Model | CLIP↑ | Aesthetic↑ | LLM-G↑ | Align.↑ | Real.↑ |
|---|---|---|---|---|---|---|
| Gaussian Splatting | | | | | | |
| MVDream | - | 24.71 | 5.73 | 4.90 | 2.74 | 2.28 |
| RewardSDS | IR | **25.24** | **5.79** | **5.31** | **4.11** | **3.13** |
| NeRF | | | | | | |
| MVDream | - | 26.19 | 5.83 | 5.86 | 3.51 | 3.14 |
| RewardSDS | IR | 27.12 | 5.97 | 6.07 | **4.21** | 3.79 |
| RewardSDS | R3D | **27.43** | **6.01** | **6.23** | 3.75 | **3.84** |
| DreamReward | - | 28.66 | 6.31 | 6.51 | 2.87 | 2.39 |
| DreamReward + Ours | R3D | 28.78 | 6.38 | 6.62 | **3.24** | **2.57** |
| DreamReward + Ours | A | **28.85** | **6.45** | **6.65** | 2.91 | 2.46 |

sampling influences DreamReward Ye et al. (2024). As DreamReward is designed to use a 3D reward model, we consider a variant where noise samples are also selected based on the same 3D reward model, but also a variant where they are selected based on a different 2D reward model (which cannot be done in DreamReward), showing our ability to align to both reward models effectively.

3DGS was trained on 30 random prompts from the DreamFusion Gallery [2], while NeRF was trained using 22 hand-crafted prompts (we refer readers to Appendix F for the full prompt list). We measure CLIPScore, Aesthetic Score, and LLM Grader as automatic metrics of 10 randomly sampled views from each generated scene and report the average score. We also consider a user study to assess alignment and realism, which follows the procedure of the 2D setting, but with a video of the 3D scene.

As shown in Tab. 3, our method consistently improves all metrics using both 3D Gaussian Splatting and NeRF backbones, demonstrating its effectiveness in enhancing text-scene alignment and overall visual quality in 3D generation. This also highlights the plug-and-play nature of our approach, as evidenced by the performance gains achieved when integrating it into the DreamReward framework. DreamReward alone achieves high automatic metric scores, where the combination of our method with DreamReward results in the best automatic metric performance. However, its user study results, assessing similar attributes (realism and alignment) are noticeably lower than ours. We attribute this to reward hacking Skalse et al. (2022), caused by direct optimization through the reward model, which leads to visual artifacts and amplifies common text-

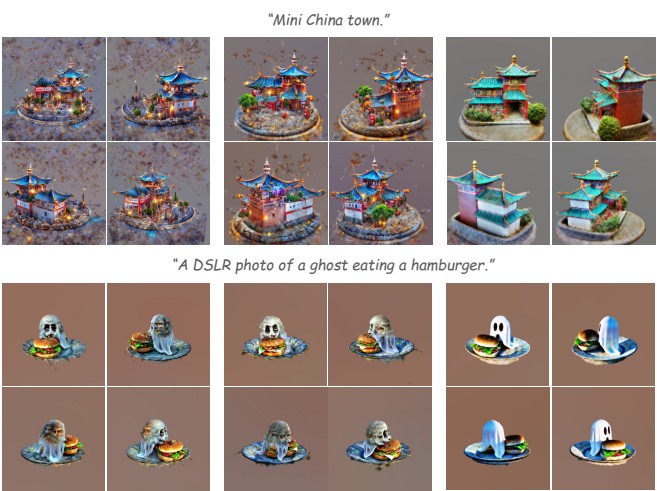

"Mini China town."

"A DSLR photo of a ghost eating a hamburger."

DreamReward     DreamReward + RewardSDS     RewardSDS

Figure 6: Qualitative comparison of DreamReward, DreamReward + RewardSDS, and RewardSDS.

to-3D issues such as the Janus problem. These flaws are often overlooked by automatic metrics but are clearly perceived in human evaluations. Based on this observation, we argue that using the reward model to guide optimization indirectly, through noise selection, is a more effective and robust approach. In our setting, the reward signal does not directly influence the loss function, but instead informs which noise samples to follow, reducing the risk of reward hacking by decoupling the reward output from gradient-based

---

[2]https://dreamfusion3d.github.io/gallery.html

optimization. Qualitative results, shown in Fig.5, illustrate the superiority of our method over the NeRF baseline. Fig.6 further provides a visual comparison with DreamReward, highlighting its flaws in comparison to RewardSDS. More visual comparisons are available in Appendix B.2 , and specifically on the attached webpage.

**Evaluating RewardSDS Across Diverse Baselines**    On the GPTEval3D benchmark Wu et al. (2024), our method achieves significant improvements over prior work, ranking at the top of the public leaderboard[3]. We compare a range of text-to-3D baselines with RewardSDS, including methods specifically designed to address known SDS issues such as over-saturation and over-smoothing Chen et al. (2023); Wang et al. (2023b); Sun et al. (2023); Qiu et al. (2024). This comparison is provided in Appendix A.1. To further demonstrate our flexibility, we qualitatively assess the effect of using different reward models in Appendix B.4, showing how RewardSDS adapts to diverse alignment objectives.

### 4.3   Image Editing

We extend the editing method introduced in DDS Hertz et al. (2023b), which can be formulated as the difference between the SDS terms for the source and target prompts. As such, our RewardDDS is simply the difference between two RewardSDS scores. We use five noise candidates and ImageReward as the reward model. We evaluate RewardDDS against DDS on 100 randomly sampled examples from the InstructPix2Pix dataset Brooks et al. (2023), pairing each target prompt with the generated image and computing automatic metrics (CLIP-Score, Aesthetic Score, and LLM Grader). Specifically, RewardDDS achieves a CLIPScore of 24.19, an Aesthetic Score of 5.80, and an LLM Grader score of 7.09, outperforming DDS, which obtains scores of 24.00, 5.77, and 6.89. A qualitative comparison is given in Fig. 7.

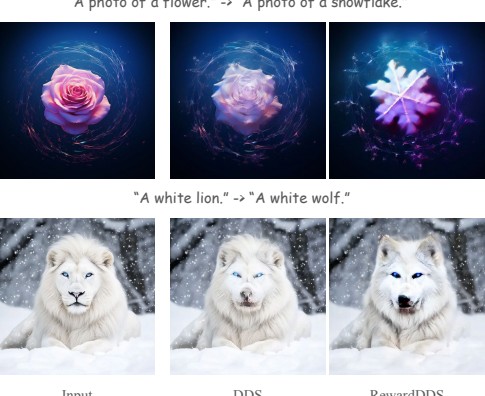

Figure 7: Qualitative comparison of image editing. We compare DDS to our adaptation, RewardDDS.

### 4.4   Ablation Studies

Our method allows for a diverse set of design choices, which we analyze here. We consider a common experimental setup for all ablations: We evaluate zero-shot text-to-image generation on 25 randomly sampled prompts from the Drawbench benchmark using ImageReward as the reward model. We use RewardSDS as our method, while the baseline is SDS.

**Noise weighting.**    We ablate the choice of noise weighting $(w^{(i)})$, applied to the $N$ candidate noise samples drawn at each iteration. Specifically, we compare the following strategies: (i) Random, where a weight of 1 is assigned to a randomly selected candidate and 0 to others; (ii) Softmax, where a softmax function is applied to the reward scores; (iii) Winner-takes-all, in which only the candidate with the highest reward is used (assigned a weight of 1, and the rest are assigned 0); (iv) Two winners-take-all (as in (iii) but with 2 highest-reward samples); (v) Step towards best, away from worst, which takes a positive step toward the candidate with the highest reward (weight 0.9) while subtracting the influence of the lowest-reward candidate (weight -0.1); (vi) Step towards top-2, away from bottom-2, which is as in (v) but with 2 highest/lowest reward samples. Tab. 4 summarizes the performance of each scheme.

Table 4: Effect of noise weighing schemes.

| Scheme | LLM-G↑ |
|--------|--------|
| (i)    | 6.69   |
| (ii)   | 7.08   |
| (iii)  | 7.05   |
| (iv)   | 7.13   |
| (v)    | 7.12   |
| (vi)   | **7.17** |

**Reward-based optimization steps.**    We now consider whether RewardSDS can be applied for a smaller number of steps $K$ out of the total number of optimization steps, while using standard SDS for the remaining steps. We vary $K$ from 0 to 1000 (out of a total of 1000 optimization steps), in increments of 100. As shown in Fig. 8 (top graph), performance improves as $K$ increases, with noticeable

---

[3]As of October 13, 2025, see `https://huggingface.co/spaces/GPTEval3D/Leaderboard_dev`.

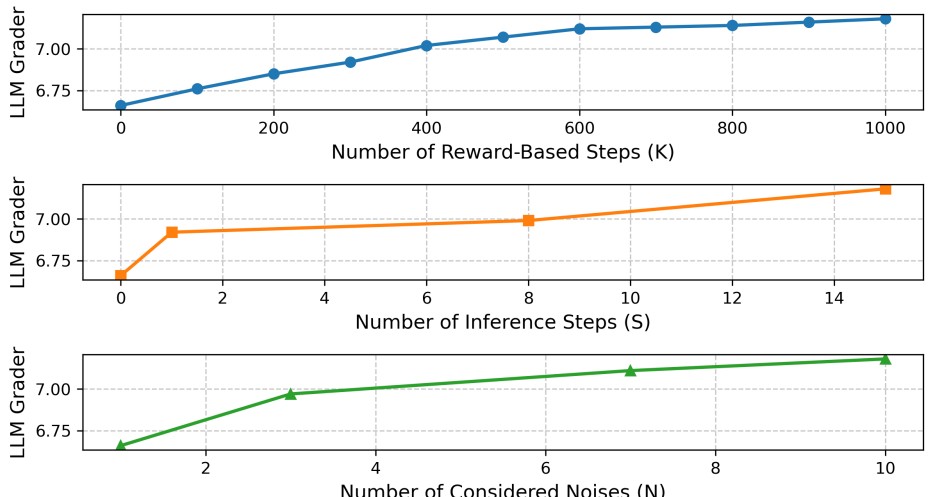

Figure 8: The top graph shows the impact of the number of reward-based steps ($K$), the middle graph presents the impact of the number of inference steps ($S$), and the bottom graph illustrates the impact of the number of considered noises ($N$).

Table 5: Comparison of model performance on text-to-image generation, for 1000 optimization steps, across different scales, evaluated on multiple metrics. Time column represents the total optimization time in seconds.

| Scale | CLIP↑ | Aesthetic↑ | LLM-G↑ | Alignment↑ | Realism↑ | Time (sec)↓ |
|---|---|---|---|---|---|---|
| Baseline | 26.94 | 5.43 | 6.87 | 2.13 | 1.98 | ∼120 |
| Small Scale | 27.20 (**8.8x**) | 5.57 (**9.6x**) | 6.99 (**8.1x**) | 3.15 (**13.2x**) | 2.42 (**11.9x**) | ∼190 |
| Medium Scale | 27.34 (2.3x) | 5.75 (3.8x) | 7.21 (4.1x) | 4.03 (4.3x) | 2.62 (3.1x) | ∼520 |
| Large Scale | **27.81** | **5.86** | **7.31** | **4.42** | **3.07** | ∼2200 |

saturation around $K \approx 600$ steps. Importantly, even for $K = 100$, we observe superior results compared to $K = 0$.

**Number of inference steps for noise selection.** We consider the effect of the number of denoising steps ($S$) used to denoise a noisy sample. We consider three different values: $S = 1$, $S = 8$, and $S = 15$, alongside a baseline without using the reward model ($S = 0$). As can be seen in Fig. 8 (middle graph), increasing $S$ consistently improves performance, with minimal increase in steps ($S = 1$) already outperforms the baseline.

**Effect of the number of noises sampled.** Next, we ablate the number of noise samples sampled (and weighted) at each optimization step, denoted as $N$ (as in Eq. 8). We test four different values of $N$: 1, 3, 7, and 10, where $N = 1$ serves as the baseline (no reward model). The results, presented in Fig. 8 (bottom graph), indicate that increasing $N$ consistently improves performance. Note the significant improvement already from $N = 1$ to $N = 3$. Qualitative results, shown in Appendix B.6 further illustrate this.

**Time vs. quality tradeoff.** By experimenting with $N$, $K$, and $S$ we observed two main conclusions: (i) increasing $N$, $K$, and $S$ improves performance, albeit at the cost of longer running time, and (ii) even small increases in these parameters already yield improvements over the baseline. Therefore, we further analyze the concrete tradeoff between quality and running time. Specifically, we evaluate four scenarios: (i) Baseline – regular SDS, i.e., $N = 1, K = 0, S = 0$. (ii) Small scale – minimal additional running time, i.e., $N = 3, K = 200, S = 1$. (iii) Medium scale – with $N = 7, K = 600, S = 8$. (iv) Large scale – with

$N = 10, K = 1000, S = 15$. All four scenarios are evaluated under the same experimental setup described above, using a total of 1000 optimization steps for text-to-image generation.

The results, presented in Tab. 5, suggest that performance consistently improves as the scales increases. Notably, even at small scale, RewardSDS yields a non-negligible improvement over the baseline despite the modest runtime overhead, achieving the best efficiency (gain per second) compared to the *larger scale*, where efficiency is computed as:

$$\text{ScorePerTimeUnit} := \frac{\text{S}_\text{Scaled} - \text{S}_\text{Baseline}}{\text{T}_\text{Scaled} - \text{T}_\text{Baseline}}$$

where $S$ and $T$ stand for Score and Time, respectively. Qualitative comparisons for all four scales are provided in Appendix B.7, further illustrating the visual improvements obtained as the scale increases.

**Additional analysis.** Beyond these ablations, we further examine the effect of gradient interpolation on image sharpness and compare RewardSDS to Score Distillation via Inversion (SDI) in Appendix A.5 and Appendix A.6, respectively. These additional studies show that RewardSDS maintains image sharpness and outperforms DDIM-based methods like SDI, highlighting its robustness and adaptability. Finally, Limitations, failure cases, and directions for future work are discussed in Appendix E.

## 5 Conclusion

To conclude, we have presented RewardSDS, a novel approach for addressing the critical challenge of aligning score distillation with ser intent through reward-weighted noise sample selection. Across extensive evaluations encompassing zero-shot text-to-image generation, text-to-3D creation, and image editing, RewardSDS consistently and significantly outperformed standard SDS and subsequent baselines on diverse metrics, including CLIPScore, Aesthetic Score, ImageReward, LLM-Grader assessments, and user studies, with models based on ImageReward and Reward3D notably demonstrating robust performance in 3D generation. The general nature of RewardSDS facilitates seamless integration with various SDS extensions and pre-trained reward models, including those that are non-differentiable, offering a flexible and scalable framework for improved alignment.

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

# A    Additional Quantitative Results

Complementary qualitative results corresponding to the experiments reported in this section are presented in Appendix B.

## A.1    Quantitative Results for GPTEval3D Benchmark

Several recent works address common issues in SDS, such as over-smoothing and color saturation. To assess the effectiveness of RewardSDS in comparison to these approaches, we evaluated it using the GPTEval3D benchmark Wu et al. (2024), which examines text-to-3D methods using GPT-4o Hurst et al. (2024) as an LLM-based judge across a diverse set of 110 prompts. This benchmark focuses on fine-grained textual and geometric alignment, using evaluation prompts and multi-view renderings and normals.

Table 6: Comparison of RewardSDS to baseline text-to-3D methods and on the GPTEval3D benchmark Wu et al. (2024). RewardSDS achieves the highest scores across most evaluation aspects.

| Method | Text-Asset Alignment↑ | 3D Plausibility↑ | Text-Geometry Alignment↑ | Texture Details↑ | Geometry Details↑ |
|---|---|---|---|---|---|
| RewardSDS | 1284.33 | **1247.93** | **1340.44** | **1383.77** | **1371.21** |
| DreamCraft3D Sun et al. (2023) | **1336.67** | 1224.95 | 1318.51 | 1373.89 | 1288.89 |
| RichDreamer Qiu et al. (2024) | 1294.85 | 1225.28 | 1259.99 | 1355.95 | 1251.28 |
| MVDream Shi et al. (2024) | 1270.55 | 1147.47 | 1250.57 | 1324.89 | 1255.46 |
| ProlificDreamer Wang et al. (2023b) | 1261.8 | 1058.73 | 1151.99 | 1246.37 | 1180.56 |
| LatentNerf Metzer et al. (2023) | 1222.33 | 1144.84 | 1156.7 | 1180.47 | 1160.77 |
| OpenLRM Hong et al. (2023) | 1202.2 | 1078.78 | 1188.83 | 1211.98 | 1173.86 |
| Fantasia3D Chen et al. (2023) | 1067.9 | 891.87 | 1005.99 | 1109.29 | 1027.48 |
| DreamFusion Poole et al. (2022) | 1000.0 | 1000.0 | 1000.0 | 1000.0 | 1000.0 |

Table 7: Extended comparison of RewardSDS and other text-to-3D methods on the GPTEval3D benchmark Wu et al. (2024), using additional evaluation metrics.

| Method | CLIP↑ | ImageReward↑ | Aesthetic Score↑ |
|---|---|---|---|
| RewardSDS | **25.7** | **-0.37** | **5.96** |
| ProlificDreamer | 25.2 | -0.51 | 5.18 |
| MVDream | 24.2 | -0.57 | 5.88 |
| LatentNerf | 25.8 | -0.42 | 5.15 |
| Fantasia3D | 22.2 | -1.39 | 4.55 |
| DreamFusion | 22.4 | -1.51 | 4.24 |

Our method outperforms prior approaches on most of these key aspects, as shown in Tab. 6, which is based on the current benchmark leaderboard [4]. In addition to the main metrics, we report extended results using complementary metrics such as CLIP similarity, ImageReward score, and Aesthetic score in Tab. 7. These metrics, also used in the main paper, provide a broader perspective on image-text alignment and perceptual quality.

---

[4]https://huggingface.co/spaces/GPTEval3D/Leaderboard_dev

Table 8: Comparison of text-guided 3D generation using standard SDS and VSD (with stable-diffusion-2.1-base), with and without reward-based sampling, over 30 randomly sampled prompts from DreamFusion gallery.

| Method | CLIPScore↑ | Aesthetic↑ | LLM Grader↑ |
|---|---|---|---|
| SDS | 21.49 | 5.34 | 3.85 |
| RewardSDS | **22.87** | **5.53** | **4.29** |
| VSD | 21.38 | 5.14 | 3.67 |
| RewardVSD | **22.03** | **5.41** | **4.11** |

## A.2 Quantitative Results for 3D Generation with 3DGS

To complement our main results in the Sec. 4.2, we provide additional quantitative comparisons for 3D generation with the 3DGS backbone Tang et al. (2023). We evaluated our reward-based method with both SDS and VSD, with stable-diffusion-2.1-base as the 2D prior, and ImageReward is the reward model. Tab. 8 provides a comparison over 30 randomly sampled prompts from DreamFusion gallery, with each score averaged over 10 randomly rendered views. These results further validate the effectiveness of RewardSDS when applied to a 3DGS-based representation.

Table 9: Comparison of SDS-Bridge and ConsistentFlowDistillation (CFD) with and without RewardSDS on zero-shot text-to-image generation.

| Method | CLIP↑ | Aesthetic↑ | LLM-G↑ |
|---|---|---|---|
| SDS-Bridge | 27.01 | 5.34 | 6.44 |
| RewardSDS-Bridge | **27.94** | **5.58** | **7.31** |
| CFD | 28.71 | 5.25 | 6.29 |
| RewardCFD | **29.43** | **5.46** | **7.41** |

## A.3 Quantitative Results for Plug-and-Play Integration

As discussed in Sec. 3, one of the main advantages of RewardSDS is its plug-and-play compatibility with existing SDS-based optimization frameworks. To demonstrate this, we apply it on top of two recent methods: SDS-Bridge McAllister et al. (2024) and Consistent Noise Distillation (CFD) Yan et al. (2025). While our primary focus is text-to-3D generation, these experiments are conducted in a text-to-2D setting to reduce computational cost. Quantitative results in Tab. 9 show consistent improvements across key metrics with RewardSDS is used as a drop-in enhancement.

## A.4 Evaluation on Counting and Spatial Relationship Prompts

While aggregate metrics such as CLIP, Aesthetic, or LLM-G provide useful quantitative signals, they often fail to capture fine-grained semantic accuracy—particularly for prompts that involve object counting or explicit spatial relations. Moreover, broad user studies can be difficult to interpret when evaluating such specific compositional behaviors. To better assess these aspects, we conducted a focused evaluation using the counting and positional categories of the DrawBench benchmark. We compared 2D generations from SDS and RewardSDS (trained with ImageReward) across all 36 prompts in these categories, which include examples such as "Three cars on the street" and "A banana on the left of an apple."

Table 10: Comparison of SDS and RewardSDS on counting and positional prompts from the DrawBench benchmark Saharia et al. (2022). RewardSDS shows improved adherence to quantitative and spatial relationships, while also achieving higher general alignment metrics.

| Method | CLIP↑ | Aesthetic↑ | LLM-G↑ | **Counting (%)↑** | **Positional (%)↑** |
|---|---|---|---|---|---|
| SDS | 26.37 | 5.46 | 6.86 | 29.4 | 31.6 |
| RewardSDS | **27.92** | **5.70** | **7.08** | **76.4** | **57.8** |

For each prompt, we examined whether (i) the number of depicted objects matched the specified count, and (ii) the positional or geometrical relationships were correctly satisfied. The quantitative results are summarized in Table 10, showing that, even though it is not a dedicated compositional generation method, RewardSDS markedly improves adherence to both counting and spatial constraints while also improving general image–prompt alignment metrics. This experiment highlights a concrete and practically meaningful advantage of RewardSDS in interpretable scenarios such as compositional or relational accuracy.

Table 11: Comparison of SDS and RewardSDS using multiple noise samples to evaluate potential effects of gradient interpolation on image sharpness and alignment. Scores are averaged across 25 generated images from identical prompts.

| Method | LLM-G (Sharpness)↑ | User Study (Sharpness)↑ | User Study (Alignment)↑ |
|---|---|---|---|
| SDS | 2.16 | 3.06 | 2.53 |
| RewardSDS (6 noises) | 2.17 | 4.04 | 3.43 |
| RewardSDS (10 noises) | **2.42** | **4.21** | **3.70** |

## A.5 Effect of Gradient Interpolation on Image Sharpness

To ensure that interpolating gradients across multiple predicted noises does not introduce blurriness, we conducted an additional analysis evaluating both quantitative and perceptual image sharpness. We compared standard SDS with RewardSDS using 6 and 10 noise samples. For each configuration, 25 images were generated from identical prompts and evaluated using two complementary measures: (i) an automated sharpness metric adapted from the LLM-G evaluation framework, scaled to 1–5; and (ii) a user study with 20 participants, who rated image sharpness and image–prompt alignment on a 0–5 Likert scale (higher is better).

Results in Tab. 11 indicate that RewardSDS maintains, and in some cases slightly improves, both perceptual and quantitative sharpness compared to SDS, even when interpolating across multiple noise samples. This suggests that the reward-guided gradient aggregation in RewardSDS preserves high-frequency details rather than averaging them out, mitigating potential over-smoothing effects.

Table 12: Comparison of SDS, SDI Lukoianov et al. (2024), and RewardSDS on NeRF-based 3D generation across 22 prompts using stable-diffusion-2.1-base as the 2D prior.

| Method | CLIP↑ | Aesthetic↑ | LLM-G↑ | ImageReward↑ |
|---|---|---|---|---|
| SDS | 23.19 | 4.53 | 3.58 | -0.83 |
| SDI | 24.06 | 4.61 | **4.29** | -1.19 |
| RewardSDS | **24.52** | **4.71** | 4.21 | **-0.59** |

### A.6 Quantitative Comparison to Score Distillation via Inversion (SDI)

As noted in Sec. 1, recent methods Lukoianov et al. (2024); Liang et al. (2024) have proposed DDIM inversion as an alternative to random noise sampling in SDS. In this approach, at each training step, a view of the 3D scene is rendered, DDIM inversion is applied up to noise level $t$, and the image is then denoised to $t-\tau$ before computing gradients with respect to the 3D representation. While this technique improves standard SDS by enforcing consistency with predicted noise, it does not allow controllability based on reward-model alignment.

To compare our method with this line of work, we optimize NeRF-based scenes across 22 prompts ( F) using the stable-diffusion-2.1-base as the 2D prior. Tab. 12 shows that our approach outperforms both SDS and the DDIM-based method (SDI), while additionally enabling alignment to arbitrary reward models.

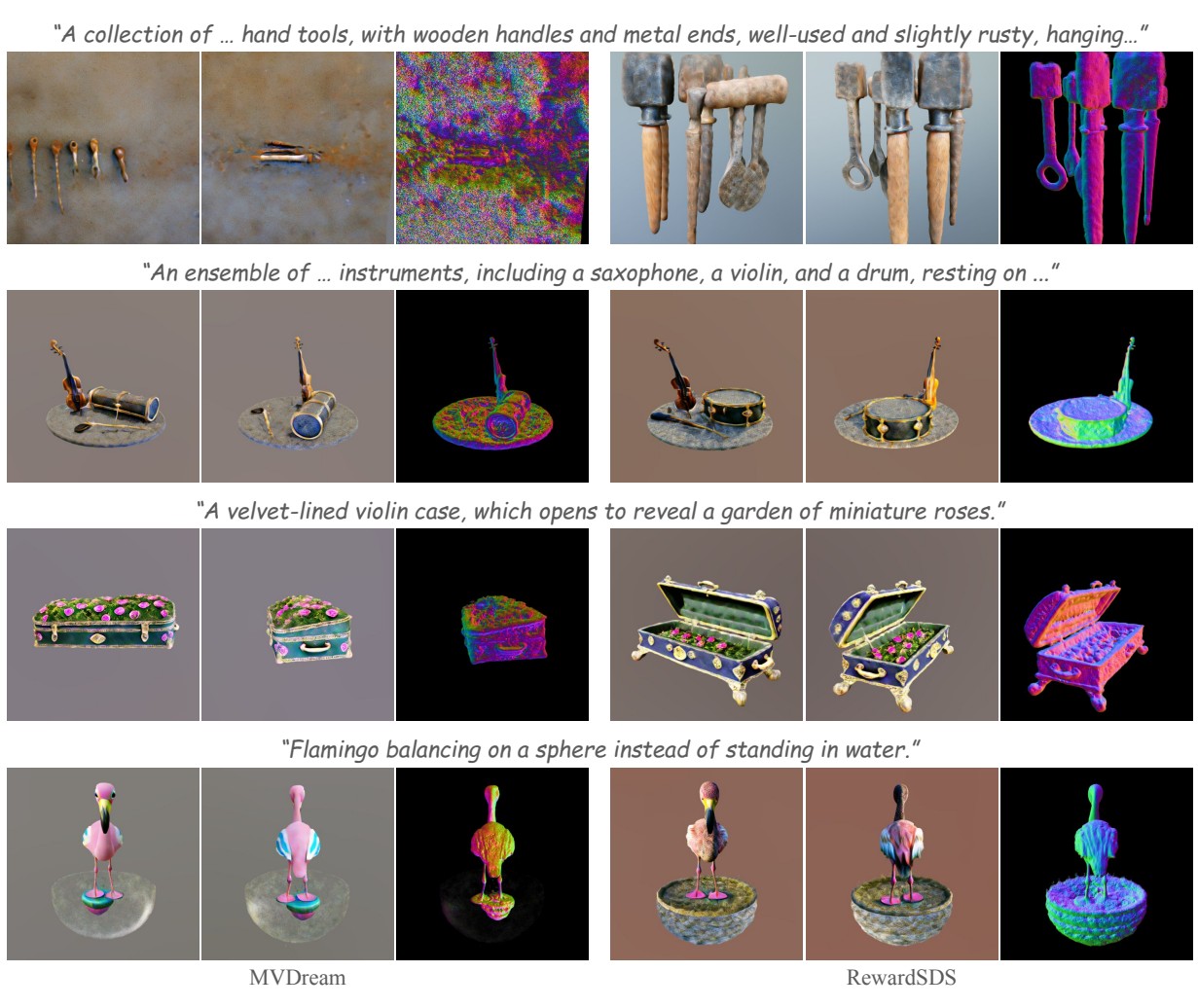

Figure 9: Qualitative comparison of RewardSDS and MVDream on challenging prompts sampled from the GPTEval3D benchmark. Wu et al. (2024)

# B  Additional Qualitative Results

## B.1  Qualitative Results for GPTEval3D Benchmark

Complementing the quantitative analysis in Sec. A.1, Fig. 9 presents qualitative comparisons between RewardSDS and MVDream on challenging prompts from the GPTEval3D benchmark. RewardSDS consistently produces more accurate and detailed results across various cases.

## B.2  Qualitative Results for 3D Generation with 3DGS

Fig. 10 presents qualitative examples corresponding to the quantitative results A.2. Additionally, Fig. 11 shows results using MVDream as the 2D prior for 3DGS optimization. These comparisons highlight the improvements in text-to-3D alignment and scene fidelity achieved by RewardSDS. The selected examples also illustrate the ability of our method to produce diverse geometry and fine-grained details, showcasing the robustness of RewardSDS in optimizing the 3DGS backbone.

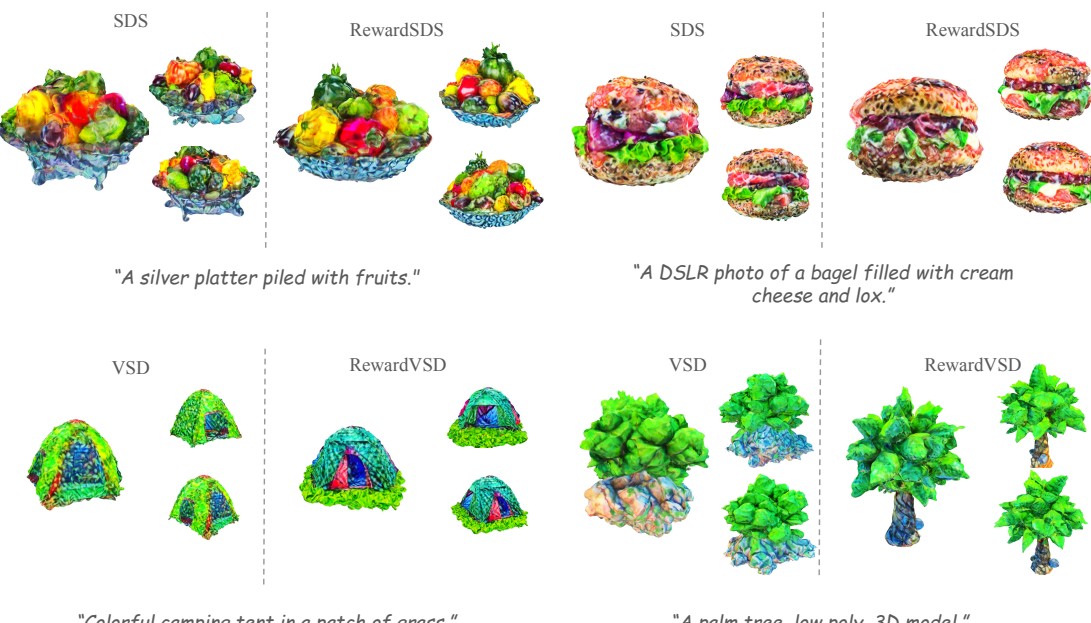

Figure 10: Qualitative results illustrating text-to-3D results for SDS compared to RewardSDS and for VSD compared to our RewardVSD (with stable-diffusion-2.1-base as our 2D prior).

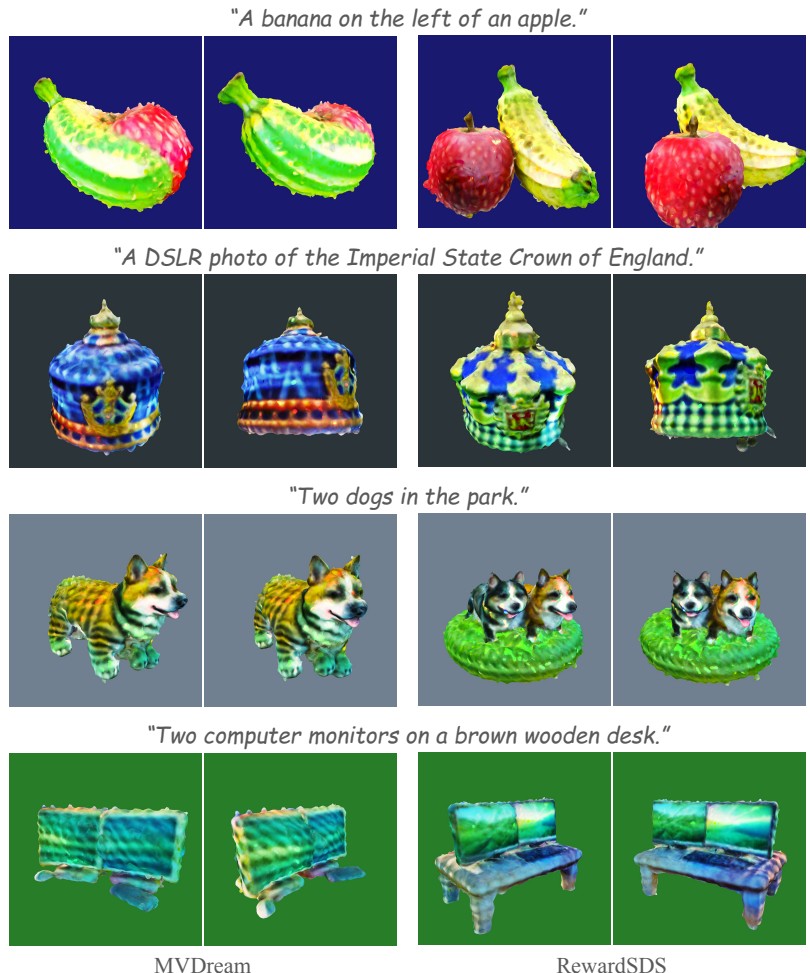

Figure 11: Qualitative comparison of text-to-3D generation based on 3DGS, in comparison to MVDream. Our method demonstrates improved alignment and visual quality, capturing more detailed geometric structures and fine-grained textures.

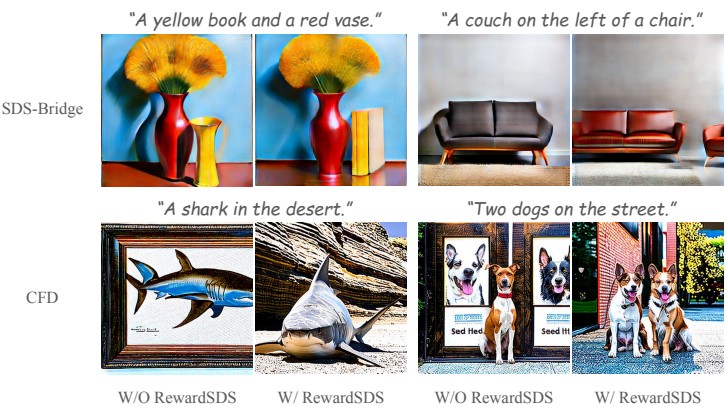

Figure 12: Qualitative comparison of SDS-Bridge and ConsistentFlowDistillation (CFD) with and without RewardSDS.

## B.3 Qualitative Examples of Plug-and-Play Integration

To illustrate RewardSDS improvements qualitatively, Figure 12 presents examples of the plug-and-play integration of reward-based noise sampling. These results further demonstrate the gains in visual fidelity and text alignment when applying RewardSDS to existing SDS-based frameworks.

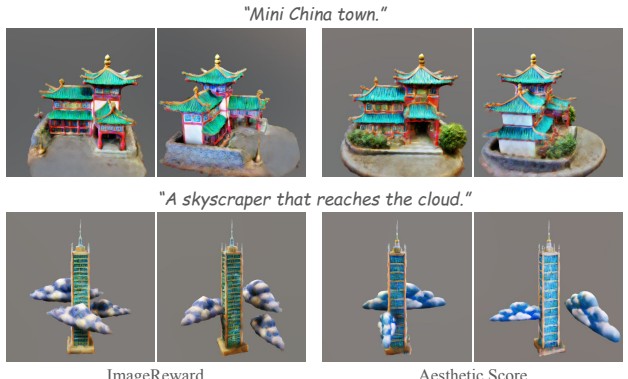

Figure 13: Qualitative comparison of text-to-3D generation using different reward models. We consider a NeRF backbone optimized with RewardSDS, either using the ImageReward reward model or Aesthetic Score reward model. As can be seen, using aesthetic reward results in adding bushes (top row) and a different (more aesthetic) colors (both rows).

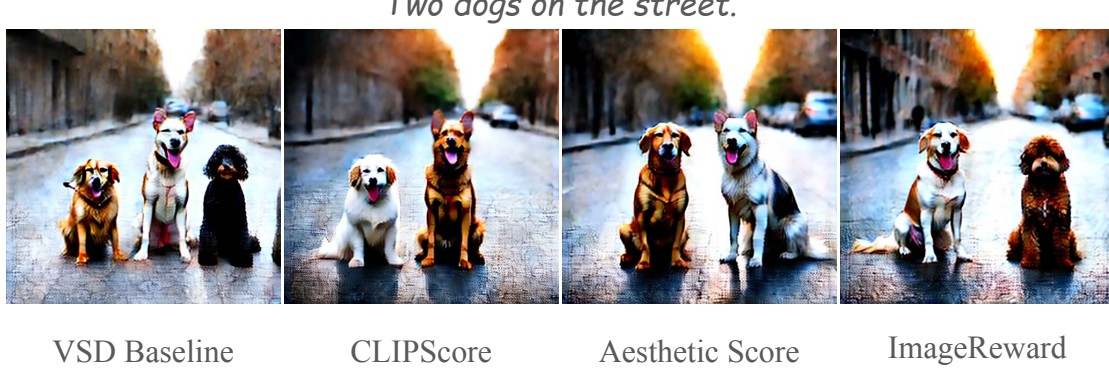

Figure 14: Qualitative comparison of generated outputs using different reward models for RewardVSD and the VSD baseline. Each row corresponds to a different reward model, with the input prompts shown at the bottom, taken from Drawbench.

## B.4 Qualitative Comparison of Different Reward Models

As a supplement to Sec. 4 of the main paper, we present additional qualitative comparisons of SDS-based generations using different reward models. First, in the 3D settings, we show results using a NeRF backbone optimized with RewardSDS, either with the ImageReward reward model or the Aesthetic Score reward model, as illustrated in Fig. 13. Fig. 14 presents 2D outputs generated using different reward models within the RewardVSD framework, alongside the VSD baseline. Together, these comparisons offer additional insight into the impact of reward model selection on the alignment and visual quality of the final generated scenes and images.

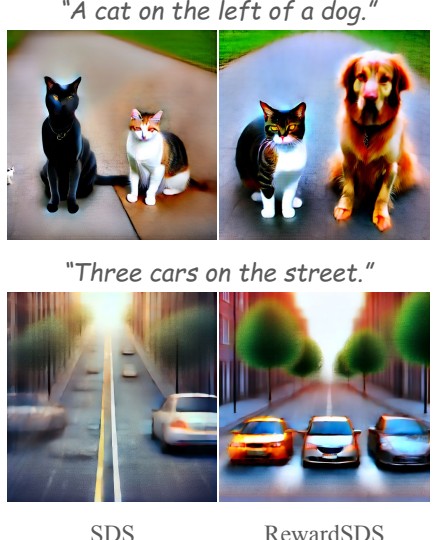

Figure 15: Qualitative comparison of SDS and RewardSDS on counting and spatial relationship prompts, showing improved alignment to object counts and spatial layouts.

## B.5   Qualitative Evaluation of Counting and Spatial Relationship prompts

Fig. 15 presents qualitative examples corresponding to the quantitative evaluation in Appendix A.4. We observe that RewardSDS produces compositions that more faithfully capture both object counts and spatial arrangements described in the prompts. These examples visually corroborate our quantitative findings, highlighting RewardSDS's advantage in fine-grained compositional alignment.

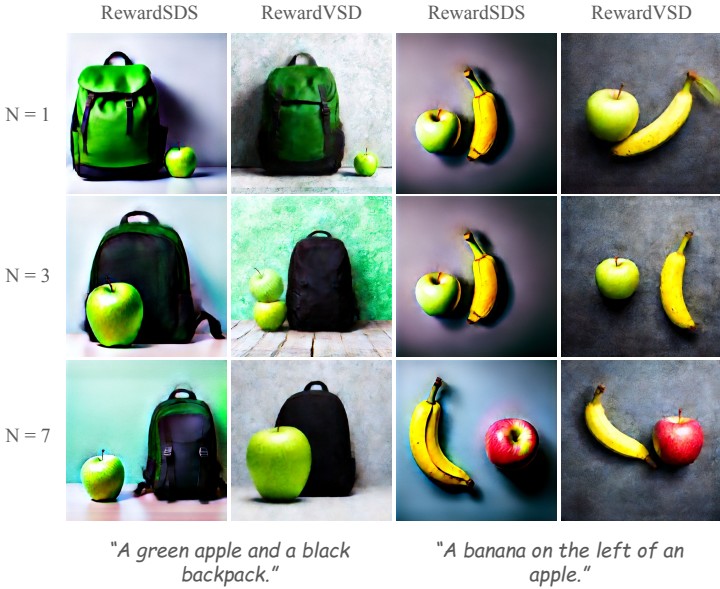

Figure 16: Qualitative results illustrating the effect of the number of considered noises ($N$). The top row presents the baseline method, while the input prompts are displayed below the images.

### B.6 Qualitative Comparison of the Effect of Number of Considered Noises

The number of noise samples ($N$) considered at each optimization step plays a crucial role in guiding the generation process. As discussed in the ablation studies, increasing $N$ leads to better alignment with the desired reward model and enhances the overall image quality. Fig. 16 provides qualitative examples illustrating how different values of $N$ affect the final generated outputs. As shown, larger values of $N$ result in more refined and coherent images, whereas smaller values introduce more variability and potential misalignment.

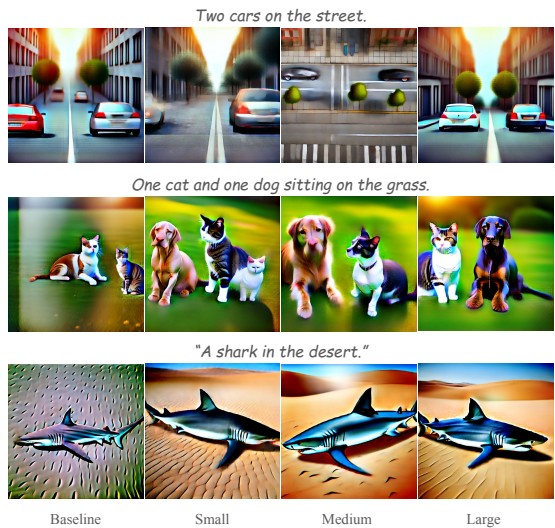

Figure 17: Qualitative comparison across different scale settings (Baseline, Small, Medium, Large). Increasing the scale produces progressively sharper, more aligned, and more coherent generations. Prompts are shown above each row.

### B.7 Qualitative Comparison Across Scale Settings

Fig. 17 provides qualitative examples complementing the scaling analysis presented in Table 5. As the scale increases from Baseline to Large, RewardSDS yields noticeably improved alignment, consistency, and visual fidelity. Even the Small scale setting shows clear gains over the baseline, supporting our claim that modest increases in $N$, $K$, and $S$ already produce substantial improvements.

## C Theoretical Analysis of RewardSDS

### C.1 Theoretical Connection of RewardSDS to DPO

In addition to the empirical results presented throughout the paper, we provide a formal derivation connecting RewardSDS to previous alignment-related works. The core idea is to optimize the parameters of a 3D scene, $\theta$, (e.g, a Neural Radiance Field) such that rendered images, when processed through a fixed pre-trained 2D diffusion model $\epsilon_\phi$ with different noise samples, align with preferences derived from a reward model $R$.

### C.1.1 Direct Preference Optimization Framework

Direct Preference Optimization (DPO) Rafailov et al. (2023) is a preference-based fine-tuning method originally proposed for language models. Given a pair of annotated samples $(x^w, x^l)$, where $x^w$ is preferred over $x^l$ for a given context $c$, DPO optimizes a contrastive loss that encourages the model to assign higher

log-probability to the preferred output:

$$\mathcal{L}_{\text{DPO}}(\phi) = -\mathbb{E}_{c,x^w,x^l} \left[ \log \sigma \left( \beta \log \frac{p_\phi(x^w|c)}{p_{\text{ref}}(x^w|c)} - \beta \log \frac{p_\phi(x^l|c)}{p_{\text{ref}}(x^l|c)} \right) \right] \tag{10}$$

where $p_\phi$ and $p_{\text{ref}}$ are the output distributions of the finetuned and reference models, respectively, and $\beta$ is hyperparameter.

For diffusion models, the DPO objective can be adapted Wallace et al. (2024) by interpreting $x^w$ and $x^l$ as two denoised predictions (e.g., from different noise samples or guidance trajectories) for the same context:

$$\mathcal{L}_{\text{D-DPO}}(\phi) = -\mathbb{E}_{c,x^w,x^l,t} \, \log \sigma \big( -\beta T\omega(\lambda_t) \big[ \, ||\epsilon^w - \epsilon_\phi(x_t^w \mid t,c)||_2^2 - ||\epsilon^w - \epsilon_{\text{ref}}(x_t^w \mid t,c)||_2^2$$
$$- (||\epsilon^l - \epsilon_\phi(x_t^l \mid t,c)||_2^2 - ||\epsilon^l - \epsilon_{\text{ref}}(x_t^l \mid t,c)||_2^2) \big] \big) \tag{11}$$

where $x_t^* = \alpha_t x_0^* + \sigma_t \epsilon^*$, $\epsilon^* \sim \mathcal{N}(0, I)$. $\lambda_t = \alpha_t^2/\sigma_t^2$ is the signal-to-noise ratio, and $\omega(\lambda_t)$ is a weighting function.

### C.1.2 Connecting RewardSDS to DPO in 3D Generation

As the original diffusion loss is closely related to the SDS lossPoole et al. (2022), our method can be seen as an extension of DPO to SDS, and is related to the above formulation. Let us consider a specific instance of the general RewardSDS formulation (see Eq.8 in the main paper), corresponding to the "step toward best, away from worst" strategy (scheme (v) in Tab.4 in the main paper). These can be interpreted as annotated examples, ranked by a reward model. Unlike the standard DPO setting, where there is a reference model, the 3D optimization setup involves a single scene being optimized. Nevertheless, we can formulate the RewardSDS loss as a reference-free DPO Meng et al. (2024) objective:

$$L_{R-SDS}(x_0 = g(\theta)) = \mathbb{E}_t \left[ w(t) \left( ||\epsilon_\phi(x_t^w, y, t) - \epsilon^w||_2^2 - ||\epsilon_\phi(x_t^l, y, t) - \epsilon^l||_2^2 \right) \right] \tag{12}$$

where $x_0$ is an rendered image of the 3D scene, $\epsilon_\phi$ is the 2D prior, $y$ is a context (e.g., text prompt), and $\epsilon^*$ denotes either the best or worst noise, corresponding to the highest, or lowest-ranked denoised image (based on scores obtained from $R$), respectively, obtained from $x_t^* = \alpha_t x_0^* + \sigma_t \epsilon^*$.

This formulation mirrors the structure of the DPO objective in Eq.11, where the model is guided to move closer to the higher-ranked (preferred) noise $\epsilon^w$ and away from the lower-ranked (less preferred) noise $\epsilon^l$, aligning gradients with preference signals. Next, let us consider the gradient of the loss:

$$\nabla_\theta L_{R-SDS}(x_0 = g(\theta)) = \mathbb{E}_t \left[ w(t) \left( (\epsilon_\phi(x_t^w, y, t) - \epsilon^w) - (\epsilon_\phi(x_t^l, y, t) - \epsilon^l) \right) \frac{\partial x_0}{\partial \theta} \right] \tag{13}$$

This update rule can be interpreted as a weighted combination of SDS gradients: a positive step in the direction of the high-reward noise and a negative step from the low-reward one. When adapting the DPO framework to optimize a 3D scene $\theta$ under a fixed diffusion prior $\epsilon_\phi$, and using preference signals derived from a reward model $R$, we arrive at this update, offering a theoretical grounding for RewardSDS strategies that emphasize high-reward noise samples while suppressing those with lower reward.

### C.2 Reward Variance Across Gaussian Noise Samples

To further understand the effect of Gaussian noise sampling on optimization outcomes, we also provide additional empirical evidence supporting why certain Gaussian noise samples lead to better outputs. To this end, we analyzed reward variance across noise samples over the course of optimization. Fig. 18 illustrates

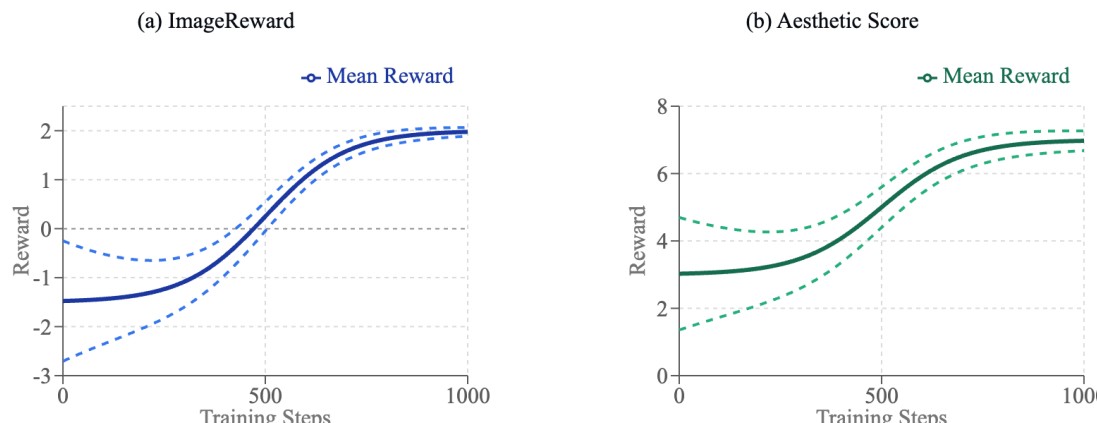

Figure 18: Evolution of reward variance across Gaussian noise samples for (a) ImageReward and (b) Aesthetic Score. ImageReward exhibits reward stabilization, while Aesthetic Score remains sensitive to subtle differences even at convergence.

the evolution of reward distributions for two representative reward models: **ImageReward** and **Aesthetic Score**.

For ImageReward, which is an unbounded reward model (with most scores in practice ranging from –3 to +3), we observed a consistent and characteristic pattern across ten prompts. The distribution of rewards follows an S-shaped trajectory over the training steps:

- In the early iterations, rewards are mostly negative, widely spread, and highly variable, with values ranging roughly from –3 to 0.

- As optimization progresses, the median and mean rewards steadily increase, passing through a transitional middle stage, where the variance remains high.

- In the final stage, rewards stabilize around positive values (close to +2), and the variance collapses to a very narrow-band (dropping from about 0.4–0.6 initially to below 0.05).

This dynamic shows that as training advances, outputs become both better aligned (higher rewards) and more consistent (lower variability).

Interestingly, for Aesthetic Score the behavior differs: while the variance decreases over time, it remains comparatively higher in the final stages, typically around 0.15 - 0.2. This is expected because prompt-alignment rewards (such as CLIP or ImageReward) focus on coarse structure early in training and converge as alignment improves, whereas rewards emphasizing fine details (such as Aesthetic Score) remain sensitive to subtle variations introduced during the final refinement steps.

This analysis highlights that the reward model itself strongly influences the variance dynamics throughout optimization.

## D   Implementation Details

In all of the experiments, including both our method and the baselines, we used a single L40s GPU. For all zero-shot text-to-image, and image editing experiments, optimization is performed using the ADAM optimizer with a learning rate of 0.01. As the text-to-image model, we employ Stable Diffusion 2.1 Base (Rombach et al., 2022b), keeping the classifier-free guidance (CFG) scale as it was in the original SDS-based method (100 for SDS, 7.5, etc.). Each 2D image is generated in 1,000 optimization steps using $N = 10$ (the number of noise samples drawn at each iteration), $K = 1000$ (the number of reward-based optimization steps used

*(a)*    *"King Kong climbing the Empire State Building."*

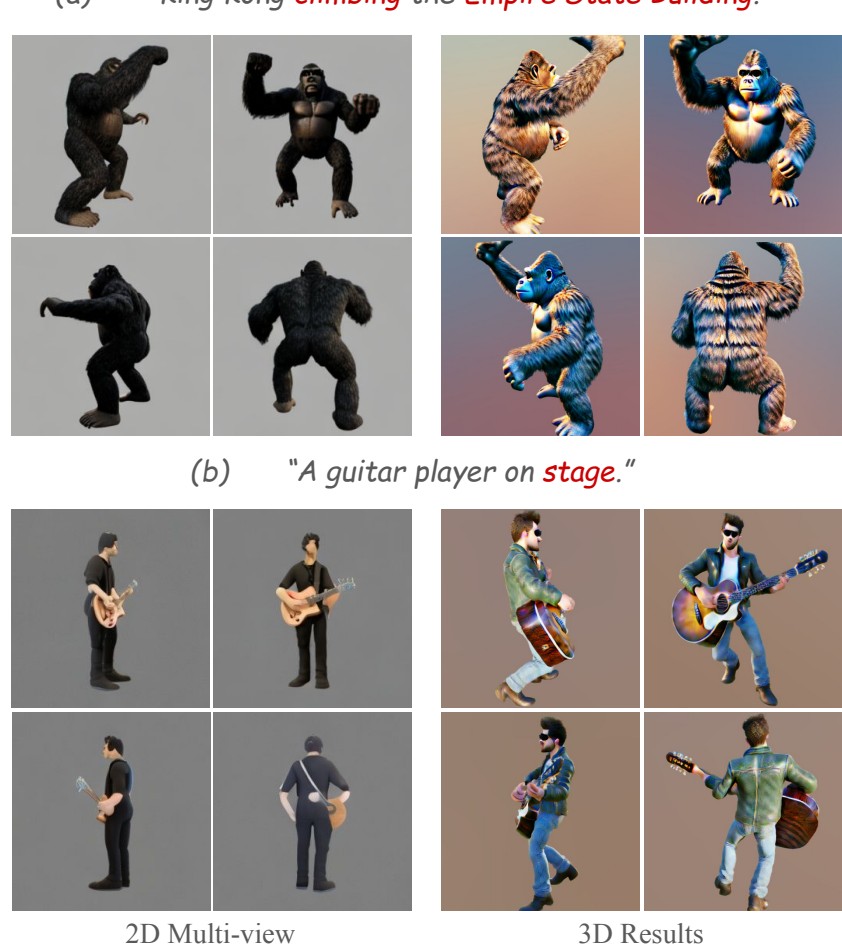

*(b)*    *"A guitar player on stage."*

2D Multi-view                    3D Results

Figure 19: In (a), fine details commonly missed by the 2D model are also absent in the resulting 3D scene. In (b), when the 2D prior (MVDream in this case) tends to generate object-centric multi-views, the resulting 3D scene struggles to complete global context such as the background.

during generation), and $S = 15$ (the number of inference steps applied during noise selection to obtain refined reward estimates). The weighting strategy assigns a weight of 0.9 to the two highest scoring samples, a weight of -0.1 to the two lowest scoring samples, and 0 to the rest. This weighting strategy corresponds to Scheme (vi) in our noise-weighting ablation (Table 4), which achieved the best overall performance. With these settings, image optimization takes 2227 seconds. For image editing experiments, generation is performed in 200 steps, taking 211 seconds per image, and we use $N = 5$, $K = 200$, and $S = 1$ as the hyperparameters, with the same weighting strategy as above. For text-guided 3D generation, MVDream Shi et al. (2024) is used as our text-to-image prior (beside the experiment described in Sec. B.2), and we employ the public implementations of DreamGaussian Tang et al. (2023) for 3DGS backbone optimization (leaving their second stage of texture refinement as is) and MVDream Guo et al. (2023) (except for the GPTEval3D experiemnts, scenes were optimized without shading) for NeRF-based optimization. We use the same settings as in the public implementations, and in NeRF training, we apply our method only in the first half of the optimization as we found that this is sufficient for convergence with our method. This choice is motivated by our ablation in Fig. 8 (top), and Fig. 18. Regarding the weighting strategy, we use the same one as in the 2D. Optimization of 3DGS takes 60 minutes and NeRF takes 7 hours.

"*Three dogs* in the park."                "*A gray* cat."

**0.803**        -0.471                **0.328**        0.023

Figure 20: In this figure, we visualize the ImageReward (↑) scores. While the reward model generally provides useful guidance, it can occasionally overlook important aspects. (Left) The model fails to accurately assess object count. (Right) The model misses stylistic attributes such as color.

# E   Limitations, Failure Cases and Future Work

This section is included to provide a more complete evaluation of our method by discussing its limitations, failure cases, and potential directions for future work.

A primary limitation of RewardSDS is the increased runtime, stemming from the denoising process performed at each SDS step to evaluate individual noise samples. This follows recent trends in generation methods that deliberately increase inference time to improve output quality Ma et al. (2025), a trade-off that is especially reasonable in 3D, where optimization is inherently slow and not designed for real-time generation, like in text or image synthesis. To address this limitation, as discussed in the Sec. 4.4, users can reduce runtime by selecting hyperparameters that better align with their computational constraints.

Additionally, although our method leverages a reward model to explore different noise samples and improve the distillation process, its performance remains constrained by the expressiveness of both the reward model and the underlying 2D prior. Despite their strong capabilities, the prior can occasionally produce suboptimal generations (see Fig. 19), and the reward model may assign inaccurate scores (see Fig. 20).

Future work can explore learning the alignment scores weights dynamically during optimization, as our current, fixed weighting was chosen through limited-scale experiments and may not generalize across prompts or optimization stages. Furthermore, evaluating noise samples using a combination of reward models, rather than a single one, may further improve alignment Eyring et al. (2024). A thorough analysis of how different reward models impact final results could offer valuable insights to users.

## E.1   Broader Societal Impacts

Our work on text-to-3D generation has potential positive applications in design, and content creation by lowering barriers to high-quality 3D asset generation. However, like other generative models, it could be misused to produce misleading or harmful 3D content. While our method is foundational and not deployed directly, we encourage responsible use and support future safeguards to mitigate potential misuse.

# F   Hand-Crafted Prompts for NeRF-Based MVDream Evaluation

As described in Sec. 4, to evaluate NeRF-based 3D generation, with and without our method, we utilized a set of 22 hand-crafted prompts. These prompts were carefully designed to be diverse and often complex, covering a wide range of objects, scenes, and subjects, thereby ensuring a comprehensive assessment of text-to-3D generation quality. The full list of prompts is provided in Tab. 13.

Table 13: List of hand-crafted prompts used to evaluate NeRF-based MVDream with and without Re-wardSDS.

| # | Prompt |
|---|--------|
| 1 | A basketball player dunking a basketball |
| 2 | A basketball player in a red jersey, high resolution, 4K |
| 3 | A bulldog wearing a black pirate hat |
| 4 | A cartoon cat eating a cheesecake, realistic |
| 5 | a DSLR photo of a ghost eating a hamburger |
| 6 | A guitar player on stage, high quality, realistic, HD, 8K |
| 7 | A man with a beard, wearing a suit, holding a pink briefcase, high quality, realistic, HD |
| 8 | A penguin with a brown bag in the snow |
| 9 | A man with a red scarf, highly detailed, 4K |
| 10 | A Shih Tzu with a bowtie, high quality, realistic, HD, 8K |
| 11 | A skyscraper that reaches the clouds, high quality, realistic |
| 12 | A tiger in the jungle, high quality, realistic, HD |
| 13 | A white sofa next to a brown wooden table |
| 14 | A young girl flying a kite, high quality |
| 15 | An astronaut riding a horse |
| 16 | Argentinian football player, celebrating a goal, HD |
| 17 | Corgi riding a rocket |
| 18 | King Kong climbing the Empire State Building |
| 19 | Mini China town, highly detailed, 8K, HD |
| 20 | Red drum set, high quality, realistic, HD, 8K |
| 21 | Two dogs in the park |
| 22 | World cup trophy, high quality, realistic, HD |

