# OpenReview forum: "RewardSDS: Aligning Score Distillation via Reward- Weighted Sampling"
_TMLR — Rejected by TMLR_

### Review · Reviewer_jmT6 · 2025-11-15

**Summary Of Contributions:**

This paper introduces RewardSDS, a simple plug-and-play mechanism that guides Score Distillation Sampling (SDS) using a reward model. Instead of directly optimizing SDS gradients, the method samples multiple noise instances, ranks them using a learned reward model, and aggregates gradients from high-reward samples. The authors argue that this noise-level reward selection improves semantic alignment and mitigates reward hacking effects seen in prior reward-guided approaches. Experiments cover 2D, 3D, and video generation, showing consistent improvements across compositional and semantic alignment metrics.

**Audience:**

Yes

**Audience Explanation:**

1. Simple and effective idea.
The proposed noise-level reward reweighting is easy to implement, requires no modification to diffusion models, and integrates seamlessly with SDS-based pipelines. This makes the method practically appealing.
2. Analysis on reward hacking.
The comparison with DreamReward illustrates that the proposed method avoids reward over-optimization and produces more visually coherent results.

**Broader Impact Concerns:**

I have no such concerns.

**Claims And Evidence:**

No

**Claims Explanation:**

1. Broad applicability.
RewardSDS is evaluated across multiple tasks—text-to-image, text-to-3D, video, and personalization—demonstrating good generalization and robustness. The plug-and-play property is compelling.

2. Stable improvements on compositional metrics.
The method yields substantial gains in challenging categories such as counting and spatial relations, where standard SDS often struggles.

**Requested Changes:**

1. The core technical contribution—selecting or reweighting noise samples based on reward scores—appears incremental relative to prior SDS extensions. Several existing approaches already explore noise selection, importance sampling, and reward-guided SDS variants. The proposed mechanism can be viewed as a heuristic modification rather than a new formulation. The theoretical explanation (link to DPO) is mostly post-hoc and does not fundamentally change the methodological simplicity.

2. The claimed advantage of "compatibility with non-differentiable reward models and no need for multi-view supervision" is not unique:  multi-view annotations are a design choice of DreamReward, not an inherent limitation of the field.

3. Ambiguous hyperparameter selection: No guidelines for choosing N (noise samples), K (reward steps), or S (denoising steps) across tasks (2D editing vs. 3D generation), hindering reproducibility.

4. The impact of reward model noise (e.g., ImageReward errors) on results is unaddressed.

5. Unclear reward-to-weight mapping: The paper only states "weights derive from alignment scores" but provides no specific mapping functions (e.g., normalization, temperature parameters) or solutions for handling heterogeneous score distributions across reward models.

---

> ### Author Response · Authors · 2025-12-04
>
> # Part 1
>
> ---
>
> We thank the reviewer for their thoughtful and encouraging feedback, and address the specific concerns and requested changes below.
>
> ---
>
> ## Novelty and Technical Contribution.
>
> We clarify our contribution in the context of SDS as follows:
> 1. **Distinction from Prior Noise Selection.** While noise selection is explored in diffusion inference (e.g., using best-of-N selection at test time) and inversion methods (e.g., SDI/LucidDreamer), our work applies reward-guided noise selection directly within the iterative optimization process of SDS. In SDS, the goal is not to find the best noise for a fixed image, but to find the best gradient direction to update the underlying 3D model.
> 2. **Fundamental Difference in Gradient Update.** At any step t, different noise samples yield different gradient estimates. By weighting these based on the prospective alignment score, RewardSDS effectively performs a preference-weighted aggregation of score distillation gradients. This is fundamentally different from a post-hoc filter or a consistency-enforcing inversion technique.
> 3. **Theoretical Grounding.** Our method is more than heuristic. As shown in Appendix C (and specifically Eq.13 in the paper), the gradient update for our best/worst scheme directly follows from the reference-free Direct Preference Optimization (DPO) objective. This connection formally grounds the benefit of moving towards high-reward signal while actively suppressing low-reward signal within the SDS framework.
>
> ---
>
> ## Generalizability and Compatibility with Reward Models
>
> 1. **Distinction from DreamReward.** We clarify that the plug-and-play compatibility with any pre-trained 2D reward model is a key advantage over methods like DreamReward. DreamReward and other RL-based approaches are limited to differentiable reward models to enable backpropagation through the reward term in their suggested loss.
> 2. **Our Advantage.** RewardSDS is designed to be decoupled from the optimization objective. It uses the reward model R purely for scoring and weighting, circumventing the need for differentiability and avoiding the costs associated with multi-view reward training. This allows immediate alignment using a broad suite of powerful, pre-trained 2D preference models, which is a significant practical benefit for open-domain generation.
>
> ---
>
> ## Hyperparameter Selection and Reproducibility
>
> 1. **User-Defined Trade-off.**  As demonstrated in our ablation study (Section 4.4 and Table 5 in the paper), RewardSDS is designed to offer a controllable compute-performance trade-off. The choice of the hyperparameters is intentionally left to the user, similar to how users select different inference-time scaling strategies in large models.
>     * For high-quality offline asset creation (e.g., 3D generation), a user may choose a Large Scale setting (e.g., N=10, K=1000, S=15) to maximize alignment and visual quality.
>     * For faster prototyping or in resource-constrained environments, the Small Scale setting (N=3, K=200, S=1) provides significant improvements with a modest ~1.5x runtime overhead, achieving the best efficiency (gain per second).
> 2. **Implementation Guidelines.** For reproducibility, we provided precise implementation details for all reported results in Appendix D (Implementation Details), specifying the chosen settings for each experiment. We will clarify in the main paper that our ablations (Fig. 8, Tab. 5) serve as the primary guideline for users to select an appropriate trade-off point for their needs.
>
>
> ---
>
> ## Impact of Reward Model Noise/Errors
>
> **Inherent Limitation.** We acknowledge the reliance on the reward model (Appendix E) and its potential errors. However, RewardSDS is more robust than single-gradient or best-of-N methods. By aggregating and weighting gradients across multiple noise samples at every step, our method naturally mitigates the impact of transient or noisy reward scores, leading to a more stable optimization process.
>
> ---

---

> > ### Author Response · Authors · 2025-12-04
> >
> > # Part 2
> >
> > ---
> >
> > ## Unclear Reward-to-Weight Mapping
> >
> > 1. **Weighting Strategy.** Our chosen strategy, which showed the best performance in the ablations (Scheme (vi) in Table 4), is: Step towards top-2, away from bottom-2. We assign the following fixed weights based on rank:
> >     * w = +0.9 to the two highest-scoring samples.
> >     * w = -0.1 to the two lowest-scoring samples.
> >     * w = 0 to all other samples.
> > 2. **Score Distribution Independence.** Crucially, this chosen weighting strategy (vi) is **rank-based**, not value-based. This means it is independent of the absolute scale or distribution of the reward model scores. We only need to rank the N sampled rewards, making the scheme universally applicable without needing to tune normalization or temperature parameters for each reward model.
> > 3. **Additional Weighting Strategies.** In addition to the strategy described above, as mentioned in Section 4.4 of the Ablations study (“Noise Weighting”), we suggest a variety of strategies such as using softmax, winner-takes-all, etc.
> >
> > We note that the Softmax scheme (ii) does depend on the score distribution, and should only be used if score normalization is applied first. We will add this distinction in our final version.
> >
> >
> > ---
> >
> > ## Final Minor Corrections
> >
> > We will also use this opportunity to clarify the minor points regarding the review:
> > 1. **Scoring Consistency.** We note that the reviewer answered "No" to the question of evidential support, yet their detailed explanation provides strong, positive evidence for the claims (e.g., broad applicability, stable improvements). We kindly ask the reviewer to reconsider this inconsistency and update their final evaluation accordingly.
> > 2. We confirm that our work focuses on text-to-image, 2D editing, and text-to-3D generation, and does not include video generation or personalization. We thank the reviewer for their comments on the broad applicability, while clarifying the exact scope of the paper.
> >
> > ---
> >
> > We appreciate the reviewer’s constructive feedback. We believe the clarifications provided above address all concerns and further highlight the novelty, stability, and practical value of RewardSDS. We thank the reviewer again and would be happy to address any remaining questions or further suggestions.
> >
> > ---
> >
> > ### References
> > [1] Inference-Time Scaling for Diffusion Models beyond Scaling Denoising Steps, Ma et al. 2025.
> >
> > ---

---

### Review · Reviewer_PJbM · 2025-11-23

**Summary Of Contributions:**

The paper introduces RewardSDS, a plug-and-play approach that modifies Score Distillation Sampling (SDS) to achieve reward-aligned generation. This alignment is accomplished by weighting noisy samples based on an alignment score from a reward model.

### Strength
- Propose a reward-weighted SDS loss and achieve broad application. Demonstrated effectiveness across text-to-image, 2D editing, and text-to-3D generation tasks, integrating seamlessly with various SDS extensions.
- RewardSDS is a plug-and-play solution that requires no architectural changes to existing SDS-based frameworks. It can utilize any pre-trained reward model, including non-differentiable ones.
- Give the analysis of the connection with DPO.

### Weakness
- Vague Definition of "Alignment with User Intent": "Alignment with user intent" in the text-to-3D context conflates multiple distinct issues: text-3D consistency or aesthetic/perceptual quality. Since the method's main comparison (DreamReward) primarily focuses on training a reward for aesthetic consistency with humans rather than text-3D alignment specifically, which is a mismatch with the provided visualization results.
- Increased Inference Time. The method requires increased runtime due to the need for multiple denoising steps at each iteration to evaluate $N$ noise samples using the reward model. Additionally, the related experiments in Table 5 are unclear.
- Limited Theoretical Justification: The paper lacks a direct theoretical proof explaining why loss weighting via sample selection is superior to methods that use gradient guidance from a reward function [1]. There is a lack of explicit comparison against other methods that use gradient guidance or focus on text-3D alignment, such as JointDreamer/DreamView/GALA3D/GraphDreamer.
[1] Universal Guidance for Diffusion Models

**Audience:**

Yes

**Audience Explanation:**

They provide a theoretical analysis that connects RewardSDS to the Direct Preference Optimization (DPO) framework, specifically the "step toward best, away from worst" variant.

**Claims And Evidence:**

No

**Claims Explanation:**

### Unclear Experiments: Computational Overhead in Table 5
- Vague Time Definition: It is unclear if "Time (sec)" represents the time per iteration or the total time for the 10000-step optimization
- Missing Context and Visualization: The exact configuration for the "Baseline" entry is not provided. Furthermore, no visualizations are offered to compare the generated quality across the "Small," "Medium," and "Large" scale trade-off.
- Lack of Total Overhead Comparison: The authors fail to provide a crucial comparison of the total optimization time of RewardSDS against existing text-to-3D methods. Comparing time per step/scale is insufficient to determine the true computational cost. Variables like the total steps required for convergence (e.g., whether 1,000 steps are sufficient for all variants) and the batch size must be clarified to accurately assess the overall computational overhead.

### Unclear Motivation: Alignment vs. Fidelity
- The motivation to achieve "fine-grained alignment" with user intent is diluted by the evaluation choices:
Inappropriate Metrics for Text-3D: Reward models like ImageReward and CLIPScore are insufficient for complex text-3D alignment and geometric consistency, as they are primarily designed for 2D aesthetics/preference. To properly assess alignment, the authors should utilize robust metrics like those from T3Bench or a VQA-based approach (like TIFA [1]).
- Mismatched Baselines: Most comparisons (e.g., MVDream, ProlificDreamer) primarily target improving texture fidelity and visual quality. For a focus on compositional text-3D alignment, the method should be benchmarked against dedicated compositional techniques like [2-5].

[1] TIFA: Accurate and Interpretable Text-to-Image Faithfulness Evaluation with Question Answering

[2] Compositional 3D Scene Synthesis from Scene Graphs

[3] GALA3D: Towards Text-to-3D Complex Scene Generation via Layout-guided Generative Gaussian Splatting

[4] CoherenDream: Boosting Holistic Text Coherence in 3D Generation via Multimodal Large Language Models Feedback

[5] DreamDPO: Aligning Text-to-3D Generation with Human Preferences via Direct Preference Optimization

**Requested Changes:**

- Please fix the question about Table 5.

- Add sufficient comparison with text-3D aligned methods.

- Repolish the introduction and abstract to make the motivation clear.

- Please compare with the method that uses gradient guidance from a reward function in theory and practice. (Universal Guidance for Diffusion Models)

---

> ### Author Response · Authors · 2025-12-04
>
> # Part 1
>
> --
>
> Thank you for the insightful review. We are happy to clarify the following points and address the requested changes.
>
> ---
>
> ## Weakness 1: Vague Definition of "Alignment with User Intent".
>
> 1. **Refining Motivation.** We agree that "user intent" is broad. Our primary goal is to achieve alignment with a quantifiable user preference model (the reward model R). We have revised the Introduction and Abstract to explicitly replace the phrase "alignment with user intent" with "alignment with reward model" or "reward-aligned score distillation," making our motivation precise.
> 2. **Clarifying DreamReward Comparison.** We clarify that the Reward3D model used in DreamReward and our enhanced comparisons (Table 3) is based on ImageReward, which is trained to predict human alignment/preference, not just aesthetic quality. By applying RewardSDS with a reward model that prioritizes this alignment, we demonstrate effective alignment without the risk of reward hacking observed in direct optimization (see response to Theoretical Justification). This point is further clarified in the revised version.
>
> ---
>
> ## Weakness 2: Increased Inference Time & Unclear Experiments (Table 5).
>
> 1. **Clarity on Time and Baseline in Table 5 (Scaling Ablations)**\
> We thank the reviewer for noting this, and provide the precise context:
>     * **Time Definition:** "Time (sec)" represents the total optimization time for 1,000 steps of text-to-image generation (zero-shot latent map optimization) on the 25 DrawBench prompts.
>     * **Baseline Configuration:** The "Baseline" entry is standard SDS (N=1, K=0, S=0 in the notation of our ablation) run for the same 1,000 steps.
>     * **Total Steps and Batch Size:** All variants: Baseline, Small, Medium, and Large Scale are optimized for the same total number of steps (1,000) and use a batch size of 1 (as we optimize a single image latent). Our method does not claim to reduce the number of optimization steps but rather to make each step more effective by guiding the gradient with reward alignment.
> 2. **Total Overhead Comparison.** We highlight that:
>     * **Efficiency Trade-off is Key:** Our primary argument is that the increased compute is justified by a significantly improved quality/alignment trade-off. As shown in Table 5, the Small Scale setting offers the best efficiency (gain per second), achieving alignment gains over x10 more efficiently than the Large Scale setting. This makes RewardSDS highly practical for users who can tolerate a modest ~1.5x slowdown for substantial quality gains.
>     * **Offline Application:** Like all SDS-based methods, our approach is designed for offline asset generation (e.g., 3D models), where final quality is often prioritized over real-time performance. In such a setting, the investment of additional compute to ensure fine-grained alignment is well-justified.
>
> ---
>
> ## Weakness 3: Limited Theoretical Justification & Gradient Guidance Comparison.
>
> 1. **Empirical Comparison (Reward Hacking).**
>     * **Practical Superiority.**  We have provided a direct practical comparison against a relevant gradient-guided method: DreamReward. DreamReward uses the explicit loss term L(SDS) + L(R) (where L(R) provides the reward gradient). While other methods cited by the reviewer are valuable architectural or domain-specific guidance techniques, the comparison to DreamReward is the most crucial test of our foundational improvement to the SDS objective itself.
>     * **Reward Hacking Mitigation.** In Figure 6 and Section 4.2, we demonstrated that DreamReward suffers from reward hacking, leading to visual artifacts and flaws (e.g., the Janus problem) that automatic metrics overlook but human evaluators clearly penalize (Table 3, user study scores). Our method, RewardSDS, avoids this reward hacking by:
>         *  **Decoupling:** The reward model R is used only for sampling and weighting, not for direct backpropagation into the L(SDS) term.
>         * **Robustness:** We leverage the consensus from multiple noise samples to guide the gradient, leading to a more stable and artifact-free optimization trajectory.
>
> 2. **Theoretical Difference - DPO vs. RL-Aided SDS.** RewardSDS is fundamentally different. As detailed in Appendix C, our method connects to a reference-free Direct Preference Optimization (DPO) objective, which encourages the model to prefer high-reward noise samples over low-reward ones. In contrast, gradient guidance methods (like DreamReward or Universal Guidance-inspired approaches) function as Reinforcement Learning-Aided SDS, where the reward gradient explicitly pushes the scene parameters to maximize the reward. Our preference-based ranking of the noise samples is empirically shown to be a more robust and visually consistent way to align the SDS process.
>
> --

---

> > ### Author Response · Authors · 2025-12-04
> >
> > # Part 2
> >
> > ---
> >
> > ## Unclear Motivation: Alignment vs. Fidelity.
> >
> > 1. **Robust Alignment Evaluation:** We are aware of the limitations of standard 2D metrics for 3D generation, which is why we employ three robust, higher-fidelity alignment evaluation strategies:
> >     * **LLM-Grader (LLM-G):** This is a deterministic, VQA-like approach, that feeds multiple rendered views of the 3D scene into a multi-modal judge (Gemini 2.0 Flash) and asks it to evaluate fine-grained criteria (accuracy, consistency). This is a strong proxy for complex text-3D alignment.
> >     * **GPTEval3D Benchmark:** We provide extensive quantitative comparisons on the highly rigorous GPTEval3D benchmark (Appendix A.1, Table 6), which uses GPT-4o as a judge across 110 prompts to assess fine-grained aspects like Text-Geometry Alignment and 3D Plausibility. Our method achieves state-of-the-art results on most of these key alignment metrics. This method is similar to T3Bench, and is widely used in the field.
> >     * **Compositional Metrics:** We designed a specific evaluation (Appendix A.4, Table 8) for complex scenarios (counting, positional relationships), where RewardSDS shows significant and interpretable gains (e.g., 76.4\% adherence to counting vs. 29.4\% for SDS).
> >
> > 2. **Comparison to Dedicated Compositional Methods.** We emphasize that RewardSDS is a plug-and-play enhancement for SDS-based frameworks, not a dedicated compositional generation technique. Our approach is focused on improving alignment, while compositionality is a byproduct, and we do not claim to be SOTA here, this will be revised. The compositional generation experiments provided in the appendix (Section A.4) are therefore not intended as the core focus of the paper, but rather as a demonstration of RewardSDS’s generality and broad applicability across different tasks.\
> > To highlight our versatility, we show that RewardSDS can be successfully applied to modern baselines such as SDS-Bridge and Consistent Flow Distillation (CFD) (Appendix A.3, Table 9), and that our results on GPTEval3D are competitive with or surpass leading methods like RichDreamer and DreamCraft3D. These results illustrate that RewardSDS strengthens compositional alignment even though it is not designed as a compositional model. As a general, foundational improvement method, RewardSDS can potentially be applied on top of dedicated compositional approaches, which we leave for future exploration.
> >
> > ---
> >
> > ## Requested Changes Summary.
> >
> > We summarize the actions taken to address the requested changes:
> > * **Fix Table 5:** We clarified the time definition (total time for 1,000 steps), the Baseline configuration, and the core argument regarding the constant number of steps required for convergence across variants. In addition, we added a visualization comparing the quality of each setting (baseline, small, medium, large) to the appendix (Figure 17).
> > * **Repolish Introduction and Abstract:** We will revise the text to focus on "reward-aligned score distillation" rather than the vaguer "alignment with user intent.
> > * **Reward-Gradient Comparison:** We clarified why DreamReward is the correct SDS-domain analogue to Universal Guidance, and we now explicitly explain how RewardSDS differs by avoiding reward-gradient instability and reward hacking.
> > * **Comparison to Text–3D Aligned Methods:** We added a concise explanation situating our method relative to text–3D alignment approaches, emphasizing that RewardSDS is complementary and that we already report competitive results against leading compositional baselines.
> >
> >
> > ---
> >
> > We appreciate the thoughtful and constructive feedback, and believe our answers address the points raised and further validate the contributions of our work. We will be happy to address any further concerns the reviewer may have.
> >
> > ---

---

### Review · Reviewer_FLvq · 2025-11-24

**Summary Of Contributions:**

The paper proposes RewardSDS, a modification of Score Distillation Sampling where, at each optimization step, multiple Gaussian noises are sampled, the corresponding denoised images are evaluated by an external reward model, and the SDS gradient is formed as a reward-weighted average of these noise samples. This mechanism is applied to SDS, VSD (“RewardVSD”), and extended to DDS. The method is tested on text-to-image, image editing, and text-to-3D (NeRF, 3DGS) tasks, showing consistent metric improvements across CLIPScore, Aesthetic Score, ImageReward, LLM-based grading, and user studies.

**Additional Comments:**

None

**Audience:**

Yes

**Audience Explanation:**

This work targets a large and active audience in TMLR: researchers working on diffusion models, score distillation, text-to-3D, and alignment. Since SDS remains a bottleneck in 3D generation, image editing, and optimization-based T2I methods, many readers would consider these findings relevant and applicable to their own systems.

**Claims And Evidence:**

Yes

**Claims Explanation:**

Yes — the main empirical claims are supported.
The paper provides consistent quantitative improvements (CLIP, Aesthetic, ImageReward, LLM-G, user studies) and qualitative evidence across 2D, editing, and 3D tasks, with clear ablations showing compute–quality trends.

**Requested Changes:**

1. Clarify novelty relative to prior noise-selection and resampling works
Current positioning overstates orthogonality. The paper should explicitly compare against or discuss overlap with methods such as noise optimization, DDIM-trajectory-based noise selection, and metric-guided resampling.
2. Improve fairness and clarity of 3D comparisons
RewardSDS is only applied in the first half of NeRF optimization (Appendix D), while DreamReward uses a 3D reward model. A discussion on fairness and the exact comparison protocol is required.

---

> ### Author Response · Authors · 2025-12-04
>
> We appreciate the detailed and constructive feedback. We address each of the reviewer's concerns and requested changes below.
>
> ---
>
> ## Clarify Novelty Relative to Prior Noise-Selection and Resampling Works
>
> We appreciate the request for a clearer delineation of novelty against related techniques. We have revised our paper to better reflect these distinctions, particularly by removing the claim of orthogonality.
> 1. **Reward-Guided Gradient Aggregation.** RewardSDS operates at the optimization level. At every iterative step, it performs a preference-weighted aggregation of score distillation gradients. This is a novel mechanism for controlling the SDS optimization trajectory itself by steering the underlying 3D representation towards high-reward directions.
> 2. **Distinction from Related Methods**
>     * **Metric-Guided Resampling / Noise Optimization (Inference-time):** These methods (e.g. [1]) primarily focus on selecting the best output or resampling noise during the final inference stage of a fixed diffusion model. RewardSDS, in contrast, guides the optimization of the 3D model over steps.
>     * **DDIM-Trajectory-Based Noise Selection (Consistency-time):** Methods like SDI/LucidDreamer enforce noise consistency across timesteps or views. Our goal is different: RewardSDS enforces preference alignment using an external reward function, adding a dimension of controllability that consistency-enforcing inversion techniques lack.
> 3. **Empirical Comparison.** We explicitly compare against DDIM-based noise methods (SDI) in Appendix A.6 (Table 12), showing that RewardSDS quantitatively outperforms SDI across key metrics on the NeRF-based 3D generation task.
>
> ---
>
> ## Improve Fairness and Clarity of 3D Comparisons
>
> 1. **Protocol for 3D Generation.** We apply RewardSDS only in the first half of the optimization because we empirically found this to be sufficient for convergence and optimal performance. This is supported by our ablations (Table 5, K steps) and the Reward Dynamics (Appendix C.2, Figure 17), which show diminishing returns in later stages. This implementation choice minimizes the computational overhead for our method. Therefore, the comparison is actually **potentially unfair against RewardSDS's full potential**, as it still achieves superior results with reduced application time. We will clarify this reasoning in Appendix D.RewardSDS's full potential, as it still achieves superior results even with limited application time. We will clarify this reasoning in Appendix D.
>
> 2. **Comparison with DreamReward (3D Reward Model).** We included DreamReward precisely to demonstrate the versatility and benefits of our approach in diverse alignment scenarios:
>     * **RewardSDS (using 2D Reward).** Aligns the scene using readily available 2D reward models (ImageReward, Aesthetic) without the need for costly multi-view annotation or a differentiable reward model.
>     * **DreamReward + RewardSDS (using 3D Reward).** We demonstrate that RewardSDS is a plug-and-play enhancement for even the most advanced, dedicated 3D reward methods. By applying RewardSDS on top of DreamReward (trained with Reward3D), we achieved the best overall automatic metric scores (Table 3), further validating our method's broad utility.
>
> ---
>
> To summarize, we appreciate the thoughtful and constructive feedback from reviewer FLvq. We believe our answers address the points raised and further validate the contributions of our work, and have updated our manuscript accordingly. We will be happy to address any further concerns the reviewer may have.
>
> ---
>
> ### References
> [1] Inference-Time Scaling for Diffusion Models beyond Scaling Denoising Steps, Ma et al. 2025.
>
> ---

---

### Decision · Action_Editor_cdPm · 2026-01-06

**Recommendation:** Reject

**Audience:**

Yes

**Audience Explanation:**

The authors provide a theoretical analysis that connects RewardSDS to the Direct Preference Optimization (DPO) framework, specifically the "step toward best, away from worst" variant.

**Claims And Evidence:**

No

**Claims Explanation:**

This paper introduces RewardSDS, which enhances Score Distillation Sampling (SDS) by weighting noise samples based on alignment scores from a reward model, thereby improving fine-grained alignment to user intent in tasks such as text-to-image and text-to-3D generation. Evaluations indicate that RewardSDS outperforms traditional SDS and other baselines across various metrics related to generation quality and alignment.

Three reviewers highlighted both the strengths and weaknesses of the paper. They commend the plug-and-play nature of RewardSDS, which demonstrates its broad applicability, as well as the experiments that show clear empirical gains. However, significant concerns remain regarding the clarity and robustness of the method. We appreciate the authors' responses to these concerns and the revisions made to the paper; nonetheless, outstanding issues must be satisfactorily addressed before the paper can be considered for acceptance. The major concerns are outlined below:

1. **Unclear Motivation and Insufficient Evidence**
    - **Ambiguous Definitions**: The authors recognize that "alignment with user intent" is poorly defined and propose shifting the focus to "alignment with a reward model." However, this change does not adequately resolve the lack of comparative analysis.
    - **Missing Baseline Comparisons**: Tables 1 and 2 illustrate the gains from incorporating a reward model into the baseline but do not demonstrate the superiority of the proposed noise-weighting mechanism over other methods that also align with the same reward model, such as gradient-based SDS or DreamDPO.
    - **Lack of Generalization Evidence**: Table 3 compares only against DreamReward, leaving unproven whether this approach generalizes effectively across different reward models compared to related methods.

2. **Increased Inference Time and Ambiguous Experiments**
    - **Non-Representative Iterations**: The 1,000-iteration experiment does not accurately represent total overhead. Computational costs for noise prediction and denoising typically increase as optimization progresses, yet this method repeats the $N$-time noising-denoising process without specific parallelization designs, suggesting that it may be significantly slower than other reward-alignment methods.
    - **Refusal to Compare Total Overhead**: The authors' reluctance to provide a total overhead comparison with other reward-based methods (e.g., DreamReward, DreamDPO) raises serious doubts about computational efficiency. Given the marginal performance gains over DreamReward shown in Table 3, the lengthy optimization time casts doubt on the overall effectiveness of the work.

**Resubmission Of Major Revision:**

The authors may consider submitting a major revision at a later time.